# GSM-Agent: Understanding Agentic Reasoning Using Controllable Environments

## Abstract

As LLMs are increasingly deployed as agents, agentic reasoning—the ability to combine tool use, especially search, and reasoning—becomes a critical skill. However, it is hard to disentangle agentic reasoning when evaluated in complex environments and tasks. Current agent benchmarks often mix agentic reasoning with challenging math reasoning, expert-level knowledge, and other advanced capabilities. To fill this gap, we build a novel benchmark, GSM-Agent, where an LLM agent is required to solve grade-school-level reasoning problems, but is only presented with the question in the prompt without the premises that contain the necessary information to solve the task, and needs to proactively collect that information using tools. Although the original tasks are grade-school math problems, we observe that even frontier models like GPT-5 only achieve 67% accuracy. To understand and analyze the agentic reasoning patterns, we propose the concept of *agentic reasoning graph*: cluster the environment's document embeddings into nodes, and map each tool call to its nearest node to build a reasoning path. Surprisingly, we identify that the ability to *revisit* a previously visited node, widely taken as a crucial pattern in static reasoning, is often missing for agentic reasoning for many models. Based on the insight, we propose a tool-augmented test-time scaling method to improve LLM's agentic reasoning performance by adding tools to encourage models to revisit. We expect our benchmark and the agentic reasoning framework to aid future studies of understanding and pushing the boundaries of agentic reasoning.

## 1 Introductions

Large language models (LLMs) have demonstrated remarkable performance on challenging reasoning tasks (Wei et al., 2022; Srivastava et al., 2023), from arithmetic word problems (Cobbe et al., 2021) to multi-hop question answering (Yang et al., 2018) and program synthesis (Chen et al., 2021). Most previous work focuses on reasoning tasks (Cobbe et al., 2021; Hendrycks et al., 2021; Saxton et al., 2019) that evaluate LLMs' *static reasoning* capability, where the model receives all necessary information from the prompt and conducts reasoning without external help. Yet, as LLMs are increasingly deployed as *agents* – systems that plan, use external tools, and iteratively refine their hypotheses – the form of reasoning that matters in practice gradually shifts from *static reasoning* to *agentic reasoning* that couples logical inference with decisions about what to read, what to ask next, when to verify, and how to recover from unproductive directions.

In this paper, we aim to understand to what extent strong static reasoning abilities of an LLM can be adapted to the agentic setting, and identify the key skills that may enable this. To achieve this, we aim to (1) compare a model's reasoning ability on the same or similar tasks under static and agentic settings; (2) identify the important skills that contribute to the performance gap between the two settings; (3) improve the model's skill in the agentic setting to enhance its agentic reasoning ability. The above steps bring two major challenges, and in this paper, we propose solutions to each of them.

**Challenge 1: Existing benchmarks fail to provide an apples-to-apples comparison of reasoning abilities under the two settings.**

*Solution:* To this end, we introduce GSM-Agent, a novel benchmark that transforms GSM8K problems into agentic tasks. Specifically, during dataset construction, each original problem is decomposed into a question and several premises; each premise is then converted into a context-rich

document and inserted into a database (the environment). During evaluation, the agent sees only the question and needs to use the provided tools (a `Search` tool and a `NextPage` tool) to discover the relevant documents before solving the math problem. Importantly, we can control the difficulty of the agentic task through careful construction of the database, e.g., by adding distracting documents. Across a broad suite of models, we observe substantial performance drops compared to the static setting where the question and all necessary documents are provided in the prompt. For example, a frontier model like GPT-5 loses roughly 33% absolute accuracy, whereas some models (e.g., DeepSeek-V3) lose up to 80%. The results demonstrate a clear and consistent gap between static and agentic reasoning in a clean and controllable setting.

**Challenge 2: We lack a framework to identify and quantify the core skills that contribute to agentic reasoning capability.**

*Solution:* To understand and analyze the core reasons of the performance gap between the two settings and what drives such significant differences in performance across models under agentic settings, inspired by Minegishi et al. (2025), we propose the concept of *agentic reasoning graph*: cluster the environment's document embeddings into nodes, and map each tool call (`Search` or `NextPage`) to its nearest node, yielding a discrete reasoning path. This framework allows us to label each reasoning step as *exploration* (first visit to a node), *exploitation* (staying within a node), or *revisit* (returning to a previously visited node after leaving). Our analysis reveals that the *revisit ratio* strongly correlates with the accuracy on GSM-AGENT, which indicates that revisit might be a core skill for strong agentic reasoning. Based on the insight, we propose a tool-augmented method, where we add a new tool that encourages the model to revisit, to improve LLMs' performance. Experimental results demonstrate that our tool-augmented method exhibits better performance than interaction-round scaling, which enforces agents to interact with the environment for more rounds without considering the quality of each interaction step.

We summarize our contributions as follows:

- We propose GSM-AGENT, a novel benchmark with a controllable environment for evaluating and analyzing the agentic reasoning capability of LLMs and providing a clear comparison between static and agentic reasoning.

- We introduce the concept of *agentic reasoning graph*, which induces a topology over the environment via clustering of document embeddings and maps tool-use traces to discrete paths. This yields interpretable, quantitative measures of *exploration*, *exploitation*, and *revisit* during the reasoning procedure at step resolution to facilitate analysis of agentic reasoning.

- Our analysis of reasoning patterns on agentic reasoning graphs reveals that revisit is an important reasoning skill that strongly correlates with agentic reasoning capability. Based on the insight, we propose a tool-augmented method to improve LLMs' agentic reasoning capability by encouraging revisit.

## 2 RELATED WORK

**Reasoning with incomplete information.** Multiple works have studied the ability of LLMs to look for missing information. Most relevant to our work, Li et al. (2025) evaluates the ability to ask the right question, including on a variant of GSM8K with missing information. However, their focus is on evaluating whether the model asks specific questions, rather than overall reasoning abilities. Zhou et al. (2025b) compare "passive" and "active" reasoning, similar to our "static" vs "agentic" reasoning, although they use different tasks for the two setups, while our dataset can be used in both scenarios, leading to a better apples-to-apples comparison.

**Agentic reasoning benchmarks.** Several benchmarks have recently been established for evaluating agentic reasoning capabilities of LLMs (Jimenez et al., 2023; Yao et al., 2024; Lu et al., 2024; Trivedi et al., 2024; Patil et al., 2025). In contrast to these works, our benchmark aims to provide a controllable environment that enables direct comparison of agentic reasoning with static reasoning.

**Understanding of reasoning.** Many recent works have focused on understanding the reasoning abilities of LLMs. Some focus on theoretically understanding the reasoning abilities and patterns of transformers (Zhu et al., 2024; Chen et al., 2024; Wang et al., 2025; Zhu et al., 2025; Huang

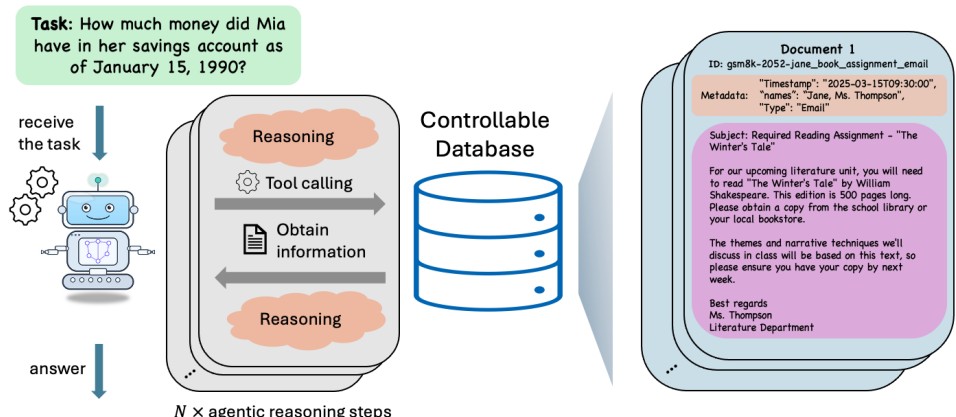

Figure 1: An overview of evaluation tasks in our GSM-AGENT benchmark. The LLM agent receives a task that only contains a question. At each *agentic reasoning* step, the agent needs to decide what information is needed, call the tool to search for information in the database, and reason about the retrieved documents. The agent also needs to decide whether all the necessary information has been collected and when to give the final answer to the task.

et al., 2025); others mainly conduct empirical studies, including the ability to self-correct (Huang et al., 2024), the reliability of reasoning on GSM8K beyond the original benchmark via synthetic extensions (Zhou et al., 2025a; Mirzadeh et al., 2025), or the reasoning behaviors or long chain-of-thought models (Yeo et al., 2025; Sun et al., 2025; Minegishi et al., 2025). Our graph-based analysis of explore, exploit, and revisit patterns was partly inspired by these works, though we extend this to the agentic setting with search tools by defining the graph through document embeddings.

## 3 GSM-AGENT BENCHMARK

In this section, we introduce GSM-AGENT, a novel benchmark with controllable environments for comprehensively evaluating the agentic reasoning capabilities of LLMs. In particular, our dataset aims to test LLM agents' abilities to combine reasoning and tool-use (mainly search) ability to solve mathematical reasoning problems by proactively interacting with the environment using tools. Below, we provide an overview of our benchmark tasks in Section 3.1, and introduce our dataset construction process in Section 3.2.

### 3.1 OVERVIEW

Our dataset $\mathcal{D} = (\mathcal{T}, \mathcal{E}, \mathcal{F})$ consists of a set of tasks $\mathcal{T}$, an environment $\mathcal{E}$, and a set of tools $\mathcal{F}$ that LLM agents can use to interact with the environment.

**Tasks.** Each task $T = (q, (p_1, \ldots, p_k), a) \in \mathcal{T}$ consists of a question $q$, $k$ premises $p_1, \ldots, p_k$ ($k$ can vary for different task instances) and the ground-truth answer $a$. Figure 2 provides an example of a task instance that consists of a question and three premises. For a grade-school-level math problem, it is easy for an advanced LLM to solve the task if all premises $p_1, \ldots, p_k$ are provided in the prompt along with the question $q$. In our benchmark, the LLM agent will only see the question $q$ without premises $p_1, p_2, \ldots p_k$ in the prompt, and it needs to use tools in $\mathcal{F}$ to find all necessary information in the environment $\mathcal{E}$ to solve the task (see Figure 1 for a pictorial illustration).

**Environments.** The environment $\mathcal{E} = \{D_1, D_2, \ldots, D_m\}$ consists of a set of documents, where each document corresponds to a premise of a task in $\mathcal{T}$. Let $g_D(\cdot)$ be a document generator, where $g_D(T) = g_D(q, (p_1, \ldots, p_k)) = (D_1, D_2, \ldots, D_k)$ and the generated document $D_i$ contains all necessary information of the premise $p_i$ for all $1 \leq i \leq k$. See the document generation part of Figure 2 for a pictorial illustration. We will introduce the details of our implementation of the document generator $g_D(\cdot)$ in Section 3.2. Since our environment $\mathcal{E}$ is a set of documents, we also call $\mathcal{E}$ a database in the rest of the paper.

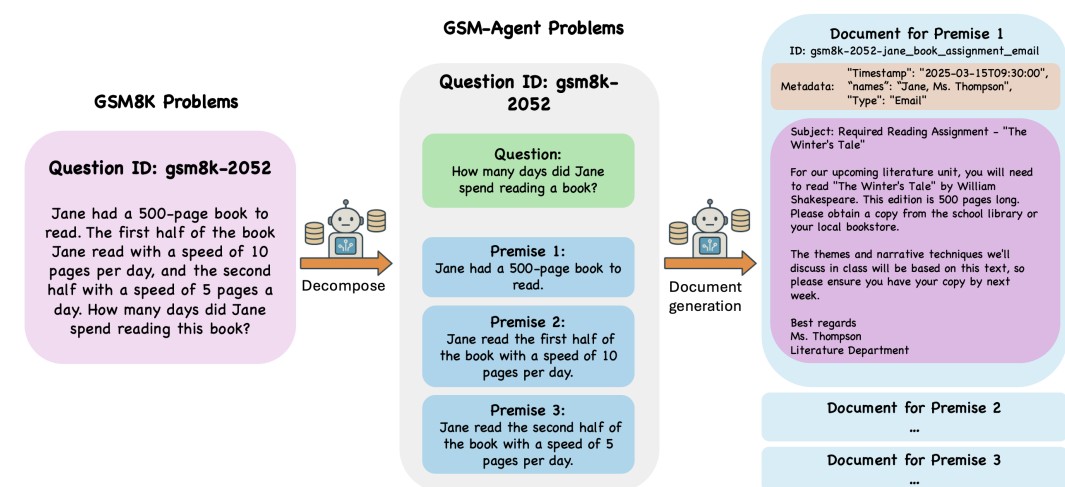

Figure 2: Data processing overview. We first decompose a GSM8k problem into a question and several premises, and then generate a document for each premise to cover its essential information.

**Tools.**    In our benchmark, we provide two tools $\mathcal{F} = \{\text{Search}(\cdot), \text{NextPage}(\cdot)\}$. For the search tool, an LLM agent can specify a query prompt $x$ (which can be of any format, such as a sentence or only several keywords) and call $\text{Search}(x)$. The search engine will return the top 5 most relevant documents in $\mathcal{E}$ to the query $x$. The agent can also use the $\text{NextPage}(\cdot)$ tool, which will return the next five most relevant documents. The LLM agent is allowed to call $\text{Search}(x)$ for multiple different query prompts $x$ that are decided by the agent itself. For each call of $\text{Search}(x)$, the agent is also allowed to call $\text{NextPage}(\cdot)$ at most 19 times for that search, resulting in retrieving up to the top 100 most relevant documents in the database. See Figure 1 for a pictorial illustration and Section 3.2 for more details about the search engine.

During evaluation, for each task $T$, an LLM agent will receive the corresponding question $q$ (without premises) in the prompt, then reason about what information is missing, call tools $\text{Search}(x)$ with appropriate search query $x$ and $\text{NextPage}(\cdot)$ to collect information from the environment $\mathcal{E}$, and solve the problem with information extracted from the retrieved document. In our dataset, the ground-truth answer $a$ is a numerical value for each task, so we directly compare the final answer given by an LLM agent $\hat{a}$ to $a$, where the task is solved iff $\hat{a} = a$.

## 3.2 DATASET CONSTRUCTION

In this section, we introduce our dataset construction procedure in detail. Our GSM-AGENT dataset is built upon the well-known GSM8k problem set, where each task $T$ in our dataset is constructed based on a problem instance in GSM8k. A simplified dataset construction pipeline is illustrated in Figure 2, while the whole pipeline consists of five stages: (1) data preprocessing; (2) problem decomposition and sharding; (3) document generation; (4) data filtering; (5) database construction. Note that some steps in our pipeline involve using LLMs to process data. Unless otherwise specified, we use `Claude-3.5-Sonnet` as our default LLM to facilitate data processing.

### 3.2.1 DATA PREPROCESSING

As shown in Figure 2, for each GSM8k problem, we decompose it into a question and several premises, and convert each premise into a document which will be added to our database. However, naively processing each problem will cause issues.

First, different problems might share the same name of the protagonist(s). Although this is not an issue in the original GSM8k problem since each problem is independent, it could cause conflict or ambiguity in our database, as documents for different tasks will be added to the same database. For example, Alice might spend 5 dollars on ice cream in document $D_1$ for one task $T_1$, and also pay 20

dollars for a book in document $D_2$ for another task $T_2$, which renders the question "How much did Alice spend in total?" ambiguous.

Second, some problems only contain a generic entity without a specific name. For example, consider a task such as "a bookshelf has 20 books at the top and 40 books at the bottom, and how many books are there in the shelf in total". While the original problem is self-contained, separating the question from premises renders the question "how many books are there in the shelf in total" again confusing and ambiguous since it is unclear which specific bookshelf the question refers to.

To address the above two issues, we carefully design the following three data preprocessing steps to disambiguate the tasks. **Step 1: Entity detection.** In this step, we use LLM to detect the main character of each problem. For problems with a generic main character without a specific name, we flag it as generic. **Step 2: Name assignment for generic entities.** In this step, we assign different names to generic entities. **Step 3: Timestamps assignment to differentiate problems sharing the same entity.** At this step, we assign different timestamps to problems sharing the same entity to ensure no conflict between documents from different problems. The details can be found in Appendix B.

### 3.2.2 PROBLEM DECOMPOSITION AND SHARDING

After systematic data preprocessing to ensure no ambiguity in our tasks and no conflicting documents in our environment, our next step is to decompose each preprocessed problem into a question $q$ and several self-contained premises $p_1, p_2, \ldots, p_k$. See the "decompose" part of Figure 2 for an example. The decomposition is executed by an LLM agent, where the agent receives a preprocessed problem along with its timestamp as the input, then breaks down the problem narrative into a list of individual self-contained and consistent premises, rephrases the core question, and carries over the timestamp. In particular, if the problem shares an entity name with another problem, its timestamp will be explicitly stated in the question $q$ outputted by the agent to make sure the question itself is unambiguous.

### 3.2.3 DOCUMENT GENERATION

The next stage is document generation. The main purpose of this stage is to convert each premise of each problem into a context-rich document, which will be added to our database (i.e., the environment $\mathcal{E}$) in the final stage. We conduct the following three steps to ensure a high-quality database and a reasonable level of difficulty for our tasks. **Step 1: Hierarchical document generation.** At this step, we generate a high-level coherent story for a problem. **Step 2: Independence verification.** We prompt Claude-3.5-Sonnet, giving it each document-premise pair and the original question, and asks it to judge whether the document contains extra information that is not covered by the premise. By doing so, we make sure that documents are independent, with no overlapping information. **Step 3: Document anonymization.** To ensure the difficulty of our benchmark, we randomly anonymize a subset of the document to avoid LLM agents from "cheating" by blindly querying the name of the main character. The details of each step can be found in Appendix B.

### 3.2.4 DATA FILTERING

Since the previous stages involve using LLM agents to process data, and original premises are converted into much longer documents, some of the generated tasks may turn out to be problematic or unsolvable. To ensure the quality of our dataset after the above data processing stages, a rule of thumb is to ensure that all generated tasks are solvable when provided with complete information (i.e., all documents corresponding to their premises). Therefore, for each problem, we test whether `claude-3.5-sonnet` can solve it given the question and its corresponding documents. We only keep the problems that `claude-3.5-sonnet` can correctly solve to ensure a high-quality dataset. After our data filtering stage, there are 7323 problems left in our dataset, with 32315 unique documents stored for the subsequent database construction.

### 3.2.5 DATABASE CONSTRUCTION

The final stage is to build the environment $\mathcal{E}$ (i.e., the database) using generated documents. We use Chroma to build our database, where the content (excluding document ID and metadata)

of each document is embedded into a vector that will be used for document retrieval. We use `text-embedding-3-large` as our default embedding model. We relegate the ablations of different embedding models to Appendix C.2.

Moreover, we built three datasets that reflect different levels of difficulty of our benchmark: **GSM-AGENT-Full**: contains all problems after data filtering; **GSM-AGENT-Medium**: contains 25% of problems after data filtering; **GSM-AGENT-Small**: contains 6.25% of problems after data filtering. For each dataset, its environment is a database that contains all the documents of problems in the dataset. For the results reported in Section 4, we evaluate models on GSM-AGENT-Full unless otherwise specified.

## 4 RESULTS & ANALYSIS

In this section, we first present the main evaluation results on a variety of mainstream LLMs on our GSM-AGENT dataset (Section 4.1), then analyze the agentic reasoning pattern and identify the core skill, which is *revisit*, that correlates to strong agentic reasoning capabilities (Section 4.2), and finally propose a tool-augmented method to improve models' agentic reasoning capability by encouraging models to revisit (Section 4.3). We use LangChain ReAct agent for all evaluation settings, with temperature set to $0.4$, max tokens $4096$, and max search rounds $50$. Each evaluation result is averaged over a $100$ subsample of the **GSM-AGENT-Full** dataset, with 3 random seeds.

### 4.1 OVERALL PERFORMANCE

Table 1: Evaluation results under zero-shot prompting with ReAct agent across models.[1] Acc and FF are shown as percentages; other metrics use the units indicated. Acronyms: Acc=Accuracy; SR=Search Rounds, which is the number of tool calls to solve a task; Dur(s)=Duration (seconds), which is the time the agent spent to solve the task; SC=Search-Complete rate, which is the proportion of tasks where the agent found all relevant documents; ER=Extra Rounds, which is the number of tool calls after all relevant documents are found; FF=Follow-Format rate, which is the proportion of tasks that an agent follows the required format to solve; PremT=Premature-Total rate, which is the proportion of tasks that an agent attempted to provide a premature answer before making the final decision; TotTok=Total Generated Tokens; Tok/R=Mean Tokens per Round. All results are averaged over three random seeds. For each metric, ↑ indicates higher is better and ↓ means lower is better.

| Setting | Acc ↑ | SR | Dur(s) | SC ↑ | ER ↓ | FF ↑ | PremT ↓ | TotTok ↓ | Tok/R ↓ |
|---|---|---|---|---|---|---|---|---|---|
| Solvable by any model | 88.00% | Nan | Nan | Nan | Nan | Nan | Nan | Nan | Nan |
| o3 | 68.46% | 13.33 | 117.85 | 53% | 4.89 | 95% | 0% | 5775.75 | 386.03 |
| GPT-5 | 66.78% | 9.98 | 116.00 | 52% | 2.18 | 100% | 1% | 7184.10 | 615.99 |
| Grok-4 | 53.00% | 7.19 | 126.01 | 42% | 2.86 | 100% | 0% | 3817.42 | 599.72 |
| Claude-4-Sonnet [2] | 56.00% | 7.39 | 42.13 | 42% | 3.14 | 100% | 4% | 731.80 | 98.23 |
| Gemini-2.5-Pro | 38.33% | 2.93 | 51.59 | 25% | 0.20 | 82% | 3% | Nan | Nan[3] |
| Kimi-K2-Instruct | 37.42% | 5.41 | 31.00 | 24% | 0.53 | 92% | 0% | 245.34 | 56.18 |
| Gemini-2.5-Flash | 25.33% | 1.88 | 17.13 | 14% | 0.12 | 99% | 4% | Nan | Nan |
| GPT-4o | 22.67% | 1.92 | 21.27 | 22% | 2.72 | 94% | 1% | 135.20 | 92.22 |
| Llama-4-Maverick | 20.00% | 2.10 | 21.94 | 17% | 0.26 | 97% | 3% | 504.93 | 211.30 |
| DeepSeek-V3 | 19.42% | 0.94 | 14.30 | 8% | 0.00 | 82% | 0% | 38.95 | 41.33 |
| Qwen3-235B | 19.30% | 1.13 | 25.76 | 19% | 4.40 | 96% | 0% | 184.82 | 173.19 |
| Llama-4-Scout | 12.54% | 2.07 | 14.93 | 9% | 1.76 | 86% | 4% | 215.48 | 118.96 |

Table 1 presents the evaluation results of different models, which show significant gaps in agent performance. The top-performing models, o3 and GPT-5 achieve accuracies of 68.46% and 66.78%, respectively. Notably, they use substantially more search rounds before reaching a conclusion, with averages of 13.33 for o3 and 9.98 for GPT-5. This suggests their ability to conduct longer-horizon

---

[1]We observe that zero-shot prompting renders the most stable result across most models, better than few-shot prompting or multi-agent systems. However, it cannot render meaningful results for DeepSeek-R1 and Claude-Opus. DeepSeek expects a different tool-use format than other models. Claude frequently asks the user for additional information and thus requires careful prompting to fix. To ensure a fair comparison across models, we excluded the two models in the evaluation.

[2]Anthropic API does not support specifying a random seed. We run the evaluation only once for Claude models.

[3]Number missing due to LangChain output format mismatch for Gemini model series.

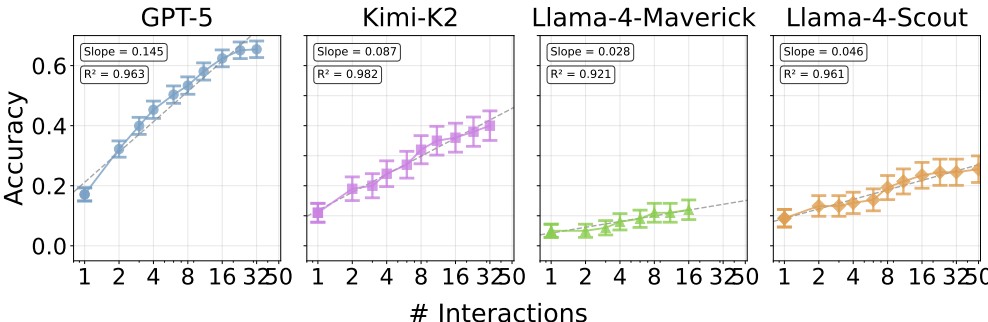

Figure 3: Interaction round scaling. The number of iteration rounds is defined as the number of tool calls. For each model, we first collect the reasoning trajectory on each task under zero-shot prompting. For a specified number of interaction rounds $n$, a trajectory is considered correct either if it answers the task correctly within $n$ rounds, or it successfully collects all necessary documents within $n$ rounds and gives a correct answer eventually. GPT-5 exhibits a much stronger interaction-round scaling than the other three models. Note that $x$-axis is in logarithmic scale.

workflows without human intervention. Following this top tier, Grok-4 also attains strong performance with an accuracy of 53.00%.

In contrast, a large group of other models exhibits considerably lower accuracy. Gemini-2.5-Pro reaches only 38.33% accuracy with an average of 2.93 search rounds, suggesting its performance gap with GPT-5 (66.78% Acc, 9.98 SR) stems primarily from fewer search rounds. We prompt all models to put the final answer after "####", top models like GPT-5 and Grok-4 demonstrate perfect (100%) format adherence, indicating robust instruction-following capabilities. Conversely, models like o3 and Gemini-2.5-Pro make more errors following this simple formatting instruction.

The large performance gap between models is puzzling, since our GSM-AGENT environment—adapted from GSM8K—requires only grade-school math and basic common sense reasoning. Nevertheless, even frontier models fall short of perfect performance, with the best accuracy reaching just 68.46% (o3), while Llama-4-Scout only achieves 12.54%. These discrepancies raise an important question:

*What drives such big differences in performance across models, given such a simple environment?*

**Simple interaction-time scaling does not improve agentic reasoning.** Table 1 shows that models tend to achieve higher accuracy when they perform more search rounds. A natural hypothesis for the observed performance gap is therefore differences in interaction-time scaling. To test this, we selected three open models (Kimi-K2-Instuct, Llama-4-Maverick, and Llama-4-Scout) and prompted them to continue searching whenever they attempted to stop. For comparison, we also measured the test-time scaling behavior of GPT-5, a representative strong proprietary model. As shown in Figure 3, the accuracy of the open models improves only marginally with additional search rounds, exhibiting far weaker interaction-time scaling than the proprietary models (GPT-5). This finding suggests that we must look beyond interaction time and instead examine the quality of interaction choices — this motivates the design of our *agentic reasoning graph*.

### 4.2 IDENTIFYING AND MEASURING CORE SKILLS CONTRIBUTING TO AGENTIC REASONING

In this section, we propose a new framework to understand and analyze models' agentic reasoning patterns and, thus, identify the core skills that contribute to a model's reasoning ability in agentic settings. Inspired by Minegishi et al. (2025), we define the *agentic reasoning graph* below.

**Nodes of the agentic reasoning graph.** Assume the environment $\mathcal{E} = \{D_i\}_{i=1}^N$ contains $N$ documents. Denote the embedding model to be $e_\theta(\cdot)$ and thus $e_i = e_\theta(D_i) \in \mathbb{R}^d$ is the embedding vector of document $D_i$. We run $K$-means ($K = 250$) on $\{e_i\}_{i=1}^N$ to get clusters $\{C_k\}_{k=1}^K$ with centroid $\{c_k\}_{k=1}^K$, and each centroids $c_k \in \mathbb{R}^d$ correspond to a node $v_k$ in the graph. Therefore, the vertex set of the agentic reasoning graph is $V = \{v_1, \ldots, v_K\}$. See Table 2 for a visualization of the vertex

(a) Visualization of Nodes

(b) Representative Nodes from The Database Graph

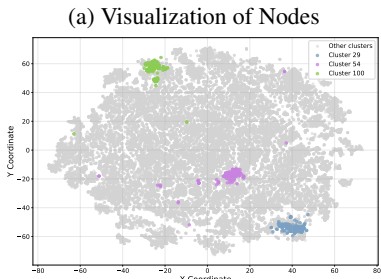

| Node | Sample Document IDs |
|---|---|
| Grocery Receipts (Node 29) | laura_flour_purchase_receipt chocolate_purchase_receipt |
| Fruit Counting (Node 54) | xena_total_fruit_count chads_apple_inventory_log |
| Youth Sports Stats (Node 100) | wario_kick_direction_analysis james_touchdown_stats |

Table 2: *Left (a)*: A t-SNE visualization of the search database embeddings. It highlights three clusters whose centroids define the embeddings for nodes 29, 54, and 100. *Right (b)*: A summary of documents for these nodes. The documents have semantic coherence within each cluster.

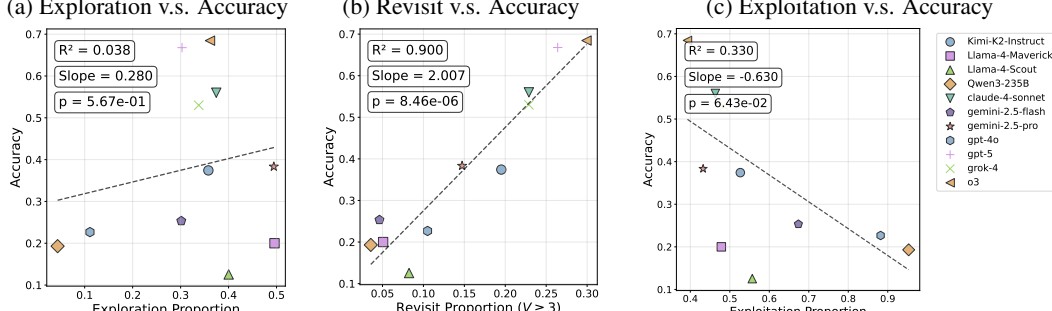

Figure 4: Correlation between accuracy and exploration, exploitation, and revisit ratio. The three ratios are defined as the proportion of exploration steps (visit a node that has never been reached), exploitation steps (visit the same node as the last step), and revisit steps (revisit a previously reached node after leaving) to the total reasoning steps. We fit linear regressions of accuracy on the three reasoning pattern ratios and report the estimated slope, $R^2$, and $p$-value for each model. The plots show that the model's accuracy has a weak correlation with the exploration ratio, a strong correlation with the revisit ratio, and a negative correlation with the exploitation ratio.

set of our database and examples for the semantic meaning of representative nodes. Throughout the main paper, we fix $K = 250$ for the $K$-means. Different choices of $K$ give aligned results. We present the ablations in Appendix C.4.

**Agentic reasoning path.** Assume the agent makes $T$ tool calls in total in the whole reasoning trace. For the $t$-th tool call, if it calls $\texttt{Search}(x)$, then the agentic reasoning node $p_t$ for the $t$-th tool call is defined as $p_t = \arg\min_{v_k \in V} \|q(x) - c_k\|_2$, where $q(x) \in \mathbb{R}^d$ is the embedding of the query prompt $x$. If the agent calls $\texttt{NextPage}(\cdot)$ for the $t$-th tool call, then the agentic reasoning node is defined as $p_t = p_{t-1}$. The agentic reasoning path is then defined as $\pi = (p_1, \ldots, p_T)$.

**Exploration, exploitation and revisit.** For each step $p_t$ in the reasoning path, we classify it into one of the three categories. For the first step $p_1$, it is always classified as an *exploration step*. For $t > 1$, if $p_t \notin \{p_1, \ldots, p_{t-1}\}$, i.e., $p_t$ has never been visited in previous steps, then $p_t$ is also an exploration step. If $p_t \in \{p_1, \ldots, p_{t-1}\}$, it is considered as an *exploitation step* if $p_t = p_{t-1}$, and otherwise a *revisit step*. The exploration ratio is defined as the proportion of exploration steps among the total number of steps $T$. Similarly, we can define the exploitation and revisit ratio. These three ratios can thus be used to quantify the reasoning pattern and facilitate deeper analysis.

Figure 4 shows that models' accuracy has a strong correlation to the revisit ratio during the reasoning trace, which implies that revisit is an important skill in agentic reasoning.

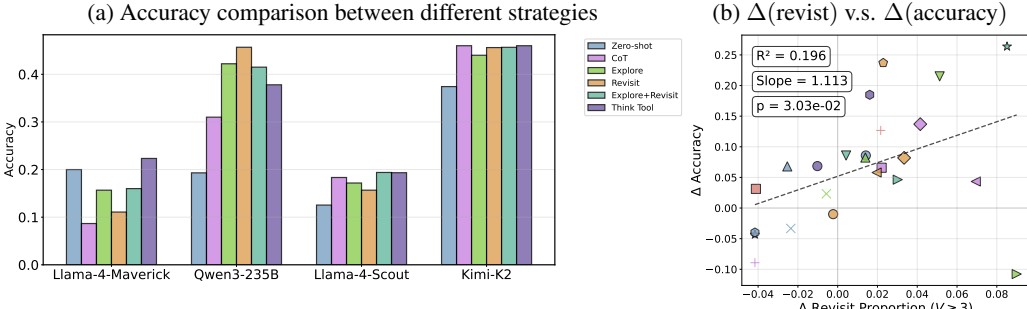

Figure 5: Visualization of performance gain via encouraging the revisit reasoning pattern. In Figure 5a, we compare five different strategies to zero-shot prompting on four different models. CoT is a prompt-only strategy where the prompt will instruct the model to think more. The remaining four strategies are all tool-augmented methods by adding different combinations of the three tools, $\text{Thinking}(\cdot), \text{Explore}(\cdot), \text{Revisit}(\cdot)$, to the tool set. For Llama-4-Maverick and Qwen3-235B, tool-augmented methods consistently outperform the prompt-based CoT strategy. For Llama-4-Scout and Kimi-K2, tool-augmented methods achieve comparable performance to the CoT method. For most cases, both the tool-augmented method and prompt-based CoT improve over zero-shot prompting. Figure 5b plots the correlation between the increase in revisit ratio and the increase in the accuracy for any of the strategies. It shows a strong correlation (with $p$-value around 0.03) between the enhancement of revisit ability and performance improvement.

## 4.3 IMPROVING AGENTIC REASONING CAPABILITY VIA TOOL-AUGMENTED SCALING

Given the insight from the analysis in Section 4.2, instead of naively scaling up the interaction rounds that is widely adopted for static reasoning, we propose to use tool-augmented scaling, which might be a more efficient scaling paradigm for agentic reasoning.

**Thinking tool, exploration tool, and revisit tool.** We introduce three new tools for our experiments. (1) $\text{Thinking}(\cdot)$ is a thinking tool, which will copy a model's preceding tokens to enforce thinking whenever called. (2) $\text{Explore}(x)$ is an exploration tool which has the same effect as $\text{Search}(x)$ while the system prompt will instruct the model to use $\text{Explore}(x)$ to explore different search queries. (3) $\text{Revisit}(x)$ is a revisit tool which has the same effect as $\text{Search}(x)$ while the system prompt will instruct the model to use $\text{Revisit}(x)$ to revisit previously called queries.

We tested four combinations of the above three tools: (1) adding $\text{Thinking}(\cdot)$ only to the tool set $\mathcal{F}$; (2) adding $\text{Explore}(\cdot)$ only; (3) $\text{Revisit}(\cdot)$ only; (4) adding both $\text{Explore}(\cdot)$ and $\text{Revisit}(\cdot)$.

Figure 5a shows that adding tools outperforms or achieves similar performance to the CoT prompt strategy in most cases. Moreover, Figure 5b shows a strong correlation between the increase of accuracy and revisit ratio, which further indicates that revisit is an important skill for agentic reasoning. The above results indicate that the tool-augmented method may serve as a more efficient test-time scaling paradigm than interaction-time scaling for agentic reasoning.

## 4.4 IMPROVING AGENTIC REASONING BEYOND THE GSM-AGENT SETTING

To show that revisit capability is an important skill in more general agentic settings beyond our proposed GSM-AGENT, we further evaluate it on DeepResearch Bench (Du et al., 2025). DeepResearch Bench is a comprehensive benchmark for deep research agents consisting of 100 PhD-level research tasks—50 in Chinese and 50 in English—covering 22 real-world fields such as science and technology, business and finance, and software (Du et al., 2025). We follow the official evaluation protocol and report RACE scores on this benchmark, and summarize our main results in Table 3.

We inject revisit behavior into the deep research agent in two ways. First, we augment the LangChain Open Deep Research multi-agent supervisor architecture (LangChain, 2025) with an explicit revisit tool available to the supervisor agent. As in our analysis in Section 4.3, the supervisor periodically

reviews the history of search queries, selects a single under-explored topic, and invokes the revisit tool to issue focused follow-up searches on that topic. For a fair comparison, we keep the total number of research iterations and the degree of concurrency identical to the original Open Deep Research configuration, so that the presence of the revisit tool is the only design difference. Second, we construct a revisit-only agent by removing the original langchain multi-agent design (summary, search workers, and supervisor) and retaining a single agent equipped solely with the revisit tool. In this variant, the main research loop degenerates into a single agent that iteratively runs search-and-revise cycles on the current draft report, using revisit-structured searches to refine the analysis.

Table 3: Comparison of performance on the DeepResearch Bench. **Bold: best results within each group**. The evaluation is repeated for three times. We use t-tests to compute the p-values between the difference of the overall scores of agents with or without the revisit tool.

| Agents | Comprehensiveness | Insight | Instruction Following | Readability | Overall Score |
|---|---|---|---|---|---|
| GPT-4.1 langchain w/ revisit | 42.94% | 39.70% | 48.60% | 46.11% | **43.85%** ($p \approx 0.01$) |
| GPT-4.1 langchain w/o revisit | 42.11% | 39.17% | 48.46% | 45.61% | 43.33% |
| GPT-5 langchain w/o revisit | 50.20% | 47.96% | 51.09% | 49.41% | 49.70% |
| GPT-5 langchain w/ revisit | 50.08% | 48.00% | 51.00% | 49.35% | 49.59% ($p \approx 0.50$) |
| GPT-5 revisit only | 51.52% | 52.82% | 51.51% | 47.39% | **51.44%** ($p \approx 10^{-17}$) |

Table 3 shows that, for both GPT-5 and GPT-4.1 within the LangChain framework, adding the revisit tool consistently improves performance or maintains comparable performance on the DeepResearch Bench. Although the absolute gains are small, paired t-tests indicate that these improvements are statistically reliable (for GPT-4.1, $p \approx 0.01$; see Table 3). Given that the gap between the Rank 1 and Rank 2 entries on the public leaderboard is only $0.01\%$, such sub-percentage improvements are nontrivial at the resolution of this benchmark. We also observed that for GPT-5, adding the revisit tool does not further improve performance, consistent with results in our GSM-AGENT benchmark: stronger models benefit less from the revisit augmenting method, since they might already have strong revisit capability or require more advanced methods to further improve it.

Moreover, the single-agent *GPT-5 revisit only* configuration achieves the highest overall score in Table 3, surpassing the more complex LangChain-based multi-agent systems. This suggests that explicitly modeling revisit behavior is itself a powerful design axis, and that strengthening structured revisit within a single strong model can be at least as effective as, and sometimes preferable to, introducing additional agents and orchestration. These findings prompt us to reconsider standard deep-research agent designs and highlight revisit capability as a first-class component rather than a minor implementation detail.

## 5 CONCLUSIONS

In this paper, we study LLMs' agentic reasoning capability, where an LLM agent needs to combine tool-use and reasoning ability to solve tasks. We first propose a novel benchmark, GSM-AGENT, where an LLM agent is required to solve grade-school math reasoning problems but must proactively search for necessary information from the environment. Our comprehensive evaluation of various models shows a significant gap in performance across different models in the seemingly simple environment. We further analyze the reasoning patterns of different models using the agentic reasoning graph and identify revisit as an important skill for agentic reasoning. Finally, we propose a tool-augmented scaling method that adds new tools to encourage the model to revisit, which improves agents' performance on our benchmark for different models. We hope that our benchmark can serve as a controllable and clean environment for future study of agentic reasoning, and our framework of agentic reasoning graph can bring new insights into better understanding and improvement of reasoning ability for LLM agents.

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

# A    DATABASE DETAILS

## A.1    STATISTICS

Let $\{D_i\}_{i=1}^{p_k}$ be the documents associated with problem $k$. We use the K-means to decide the classes of the documents $(c(D_1), \ldots, c(D_{p_k}))$. Define the "span" of problem $k$ as the number of unique classes among $(c(D_1), \ldots, c(D_{p_k}))$. We also compute the "Documents-Problem" ratio, which is the average of $p_k$. Table Table 4 displays the summary statistics among three database sizes.

Table 4: Summary statistics of the database.

| Category | Mean | Std Dev | Min | Max | Median |
|---|---|---|---|---|---|
| **Span Statistics** | | | | | |
| Full | 2.89 | 1.28 | 1 | 10 | 3.0 |
| Medium | 2.92 | 1.31 | 1 | 10 | 3.0 |
| Small | 2.84 | 1.29 | 1 | 8 | 3.0 |
| **Documents per Problem Statistics** | | | | | |
| Full | 4.41 | 1.74 | 1 | 14 | 4.0 |
| Medium | 4.41 | 1.77 | 1 | 14 | 4.0 |
| Small | 4.41 | 1.87 | 2 | 13 | 4.0 |

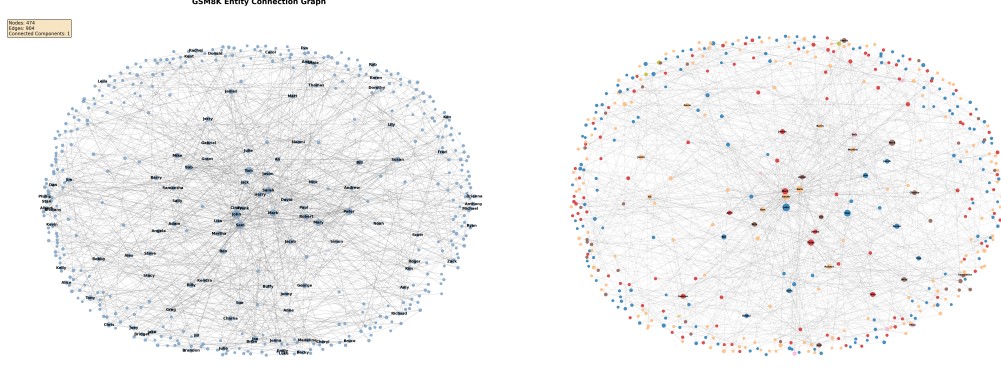

(a) Entity detection.          (b) Graph coloring for timestamp assignment.

Figure 6: Example of entity detection and timestamp assignment.

# B   ADDITIONAL DETAILS FOR DATASET CONSTRUCTION

## B.1   ADDITIONAL DETAILS FOR DATA PREPROCESSING

**Step 1: Entity detection.** First, for each original GSM8k problem, we use LLM agents to detect the name of the core narrative entity (or protagonist). The agent will prioritize identifying a single, core proper noun, such as a person's name. For example, if a problem involves both "Natalia" and "her friends", the agent will identify "Natalia" as the core entity. When there are multiple, equally important entities such as "Alice and Bob", the agent will extract both of them as the core entities. For problems where the core entity is a generic one without a specific name (such as "the girl" or "the zoo"), the agent will also extract the entity and flag it as generic.

**Step 2: Name assignment for generic entities.** For each problem flagged as having a generic core entity, we assign a name to specialize the entity. We pre-define two lists for the first name (such as ["Alice", "Ben", ...]) and last name (such as ["Smith", "Johnson", ...]), respectively. Then a full name will be systematically selected from all combinations of first and last names for each generic entity, ensuring maximum uniqueness. For entities that are not persons such as "the store", the assigned name will serve as its owner such as "John Doe's store". Finally, an LLM-based rewriting module will be invoked to rewrite the question text to naturally incorporate the assigned full name.

**Step 3: Timestamps assignment to differentiate problems sharing the same entity.** To address the issue that different problems might share the same entity names, which can result in conflict documents, we assign a timestamp to each problem. More specifically, we define an entity graph (see Figure 6a) where each node $v_i$ represents a problem $T_i$. Let $N_i$ denote the set of names of core entities in problem $T_i$. Two nodes $v_i$ and $v_j$ are connected by an edge iff they share at least one entity name, i.e., $N_i \cap N_j \neq \emptyset$. Then we assign each node $v_i$ a color $c_i$, such that no two adjacent nodes have the same color (see Figure 6b for an example of coloring the entity graph using six colors). Each color represents a timestamp, so each problem is assigned a timestamp such that each entity has different timestamps in their occurrence in different problems. The temporal separation allows the LLM agent to treat different problems as distinct events in an entity's life. To ensure the time span of a problem is reasonable, we design the timestamp to include both the year and month, such as "1990-09".

## B.2   ADDITIONAL DETAILS FOR DOCUMENT GENERATION

**Step 1: Hierarchical document generation.** To make sure the documents from the same task are consistent, we adopt a hierarchical, multi-round generation using an LLM agent as the document generator. In the first round, the agent sees the whole problem (including the question, all premises, and the timestamp) and generates a high-level, consistent story for the entire problem. In the subsequent rounds, the agent is presented with each premise one by one, and generates a single, contextually rich document that encapsulates the information from the given premise based on the high-level story generated in the first round. Each generated document contains three fields: (1) the content of the document; (2) a unique ID assigned to the document; (3) metadata of the document (timestamp, name, and type).

**Step 2: Independence verification.** To prevent information leakage between premises (i.e., one document reveals important information about another premise, especially involving quantities), which makes the task easier, we perform an independence check using an LLM agent. Specifically, for each document, the agent receives the full original problem text, the target premise that the document should cover, and a list of other premises that the document should not cover, along with the document itself. If the agent identifies information from other premises, it will propose a modification to the current document.

**Step 3: Document anonymization.** To avoid an LLM agent from "cheating" by naively searching documents only by the name of the protagonist of a given task instead of reasoning about what information is needed, we perform document anonymization, where we sample a random subset of all documents (we choose 30%), and ask an LLM agent to rewrite each document such that the name of the entity will not be presented in the content of the document. We move the entity name to the

document's metadata, and an LLM agent being evaluated can view the metadata after it successfully retrieves the document to validate whether the document is related to the given task.

# C ABLATION EXPERIMENTS FOR EVALUATIONS AND DATA CONSTRUCTION

## C.1 FULL RESULTS OF GRAPH METRICS ACROSS ALL SETTINGS

## C.2 ABLATIONS ON THE EMBEDDING MODEL AND THE DATABASE SIZE.

**Ablation on Embedding Models.**    Table 7 reports the results of our ablation study comparing different embedding functions used in the evaluation and data construction pipeline. We consider three alternatives: `text-embedding-3-large` (OpenAI, default), `text-embedding-3-small` (OpenAI), and `all-MiniLM-L6-v2` (MiniLM). Across all model families (O3, GPT-4o, Grok, Kimi, Gemini, and Llama), we observe that the choice of embedding function has only marginal impact on the reported metrics. Accuracy and follow-format rates remain very close across different embeddings, and other metrics such as search rounds, duration, and search-complete rate exhibit only minor variations. Importantly, the relative ordering of model performance is preserved regardless of the embedding choice. These findings suggest that our evaluation pipeline is robust to the specific embedding function employed, and the default choice of `text-embedding-3-large` is primarily motivated by consistency rather than necessity.

**Ablation on Database Sizes.**    Table 8 reports the results of our ablation study comparing different database sizes used in the evaluation pipeline. We consider three settings: full database (`-zeroshot`, default), medium-sized database (`-medium`, 1/4 of full), and small database (`-small`, 1/4 of medium or 1/16 of full). Across all model families (O3, GPT-4o, Grok, Gemini, Kimi, and Llama), we observe a consistent trend: performance uniformly increases as database size decreases. For instance, O3's accuracy improves from 68% (full) to 81% (medium) to 80% (small), while Grok-4's accuracy rises from 53% to 61% to 69% across the same sizes. Similarly, search-complete rates increase with smaller databases, indicating that models more successfully locate relevant documents when the search space is reduced. These findings demonstrate that the task difficulty is directly tied to database scale—smaller databases make information retrieval substantially easier. Consequently, our default full database setting provides a more challenging and realistic evaluation scenario, better reflecting the complexity of real-world knowledge-intensive tasks.

## C.3 ADDITIONAL PLOTS

We first show additional interaction-time-scaling plots across different models. All models are evaluated with LangChain, with zero-shot prompt, temperature 0.4, and max tokens 4096. Figure 7 presents the result. In general, proprietary models show better interaction-time-scaling compared with open-sourced models.

Table 5: Full results across models (all settings). Acc and FF are percentages; other metrics use indicated units. Acronyms: Acc=Accuracy; SR=Search Rounds; Dur(s)=Duration (seconds); SC=Search-Complete rate; ER=Extra Rounds; FF=Follow-Format rate; PremT=Premature-Total rate; TotTok=Total Generated Tokens; Tok/R=Mean Tokens per Round.

| setting_id | Acc | SR | Dur(s) | SC | ER | FF | PremT | TotTok | Tok/R |
|---|---|---|---|---|---|---|---|---|---|
| **Anthropic** | | | | | | | | | |
| claude-4-sonnet-fewshot | 51.5% | 9.27 | 47.9 | 0.4 | 4.41 | 100% | 0.1 | 1028.52 | 118.65 |
| claude-4-sonnet-zeroshot | 56.00% | 7.39 | 42.13 | 42% | 3.14 | 100% | 4% | 731.80 | 98.23 |
| claude-opus-zeroshot | 4.0% | 0.49 | 20.33 | 0.05 | 0.64 | 31% | 0.04 | 139.24 | 321.76 |
| **DeepSeek** | | | | | | | | | |
| DeepSeek-R1-fewshot | 0.0% | 0.0 | 32.03 | 0.0 | 0.0 | 100% | 0.0 | 0.0 | 0.0 |
| DeepSeek-R1-explore-revisit | 0.37% | 0.02 | 30.86 | 0.01 | 0.0 | 26% | 0.0 | 14.06 | 591.9 |
| DeepSeek-R1-think_tool | 0.89% | 0.0 | 47.52 | 0.0 | 0.0 | 46% | 0.0 | 0.0 | 0.0 |
| DeepSeek-R1-zeroshot | 0.31% | 0.0 | 42.59 | 0.0 | 0.0 | 55% | 0.0 | 0.0 | 0.0 |
| DeepSeek-V3-fewshot | 8.0% | 0.99 | 20.93 | 0.05 | 0.2 | 100% | 0.0 | 336.16 | 336.13 |
| DeepSeek-V3-explore | 28.41% | 1.45 | 18.58 | 0.2 | 0.11 | 97% | 0.0 | 203.42 | 134.16 |
| DeepSeek-V3-explore-revisit | 41.67% | 1.51 | 16.25 | 0.32 | 0.19 | 92% | 0.0 | 243.6 | 145.44 |
| DeepSeek-V3-revisit | 28.07% | 1.47 | 17.45 | 0.21 | 0.0 | 98% | 0.0 | 174.18 | 102.69 |
| DeepSeek-V3-think_tool | 16.67% | 2.0 | 15.03 | 0.33 | 0.5 | 100% | 0.0 | 136.0 | 86.75 |
| DeepSeek-V3-zeroshot | 19.42% | 0.94 | 14.3 | 0.08 | 0.0 | 82% | 0.0 | 38.95 | 41.33 |
| **Google** | | | | | | | | | |
| gemini-2.5-flash-fewshot | 23.0% | 2.76 | 13.34 | 0.17 | 0.12 | 100% | 0.02 | 0.0 | 0.0 |
| gemini-2.5-flash-zeroshot | 25.33% | 1.88 | 17.13 | 0.14 | 0.12 | 99% | 0.04 | 0.0 | 0.0 |
| gemini-2.5-pro-fewshot | 36.54% | 3.98 | 44.93 | 0.24 | 0.17 | 100% | 0.04 | 0.0 | 0.0 |
| gemini-2.5-pro-zeroshot | 38.33% | 2.93 | 51.59 | 0.25 | 0.2 | 82% | 0.03 | 0.0 | 0.0 |
| **Grok** | | | | | | | | | |
| grok-4-fewshot | 56.33% | 11.19 | 143.6 | 0.4 | 5.3 | 100% | 0.0 | 4936.64 | 497.62 |
| grok-4-zeroshot | 53.0% | 7.19 | 126.01 | 0.42 | 2.86 | 100% | 0.0 | 3817.42 | 599.72 |
| **Kimi** | | | | | | | | | |
| Kimi-K2-Instruct-fewshot | 42.0% | 7.85 | 29.71 | 0.26 | 1.19 | 100% | 0.0 | 406.08 | 62.58 |
| Kimi-K2-Instruct-cot | 46.0% | 6.7 | 61.99 | 0.28 | 1.15 | 98% | 0.0 | 388.96 | 70.74 |
| Kimi-K2-Instruct-explore | 44.0% | 5.95 | 70.4 | 0.31 | 1.0 | 99% | 0.0 | 555.45 | 105.71 |
| Kimi-K2-Instruct-explore-revisit | 45.67% | 6.43 | 69.19 | 0.32 | 1.0 | 99% | 0.0 | 608.29 | 104.19 |
| Kimi-K2-Instruct-interaction_scaling | 40.0% | 24.82 | 252.16 | 0.44 | 18.14 | 95% | 1.26 | 4072.78 | 278.39 |
| Kimi-K2-Instruct-revisit | 45.61% | 6.95 | 63.9 | 0.31 | 0.66 | 98% | 0.0 | 472.88 | 86.01 |
| Kimi-K2-Instruct-think_tool | 46.0% | 5.19 | 76.23 | 0.29 | 0.53 | 99% | 0.0 | 792.39 | 188.17 |
| Kimi-K2-Instruct-zeroshot | 37.42% | 5.41 | 31.0 | 0.24 | 0.53 | 92% | 0.0 | 245.34 | 56.18 |
| **Llama** | | | | | | | | | |
| Llama-4-Maverick-fewshot | 25.25% | 3.51 | 50.13 | 0.14 | 0.43 | 100% | 0.11 | 1140.33 | 383.96 |
| Llama-4-Maverick-cot | 8.67% | 1.01 | 16.25 | 0.07 | 0.0 | 16% | 0.0 | 39.94 | 39.8 |
| Llama-4-Maverick-explore | 15.67% | 1.76 | 20.07 | 0.12 | 0.54 | 30% | 0.01 | 219.27 | 115.62 |
| Llama-4-Maverick-explore-revisit | 16.0% | 1.8 | 19.25 | 0.14 | 0.56 | 30% | 0.0 | 198.43 | 91.56 |
| Llama-4-Maverick-interaction_scaling | 13.0% | 6.15 | 141.46 | 0.32 | 13.41 | 93% | 1.91 | 5300.91 | 2470.58 |
| Llama-4-Maverick-revisit | 11.07% | 1.5 | 18.65 | 0.08 | 0.32 | 25% | 0.0 | 134.74 | 69.89 |
| Llama-4-Maverick-think_tool | 22.33% | 1.73 | 26.01 | 0.15 | 0.07 | 68% | 0.0 | 595.35 | 346.51 |
| Llama-4-Maverick-zeroshot | 20.0% | 2.1 | 21.94 | 0.17 | 0.26 | 97% | 0.03 | 504.93 | 211.3 |
| Llama-4-Scout-fewshot | 13.0% | 2.05 | 9.32 | 0.07 | 0.0 | 100% | 0.02 | 313.3 | 166.5 |
| Llama-4-Scout-cot | 18.33% | 2.2 | 27.24 | 0.11 | 1.12 | 93% | 0.0 | 219.35 | 100.16 |
| Llama-4-Scout-explore | 17.17% | 2.13 | 35.36 | 0.12 | 0.28 | 86% | 0.0 | 265.44 | 116.9 |
| Llama-4-Scout-explore-revisit | 19.39% | 2.5 | 29.23 | 0.15 | 0.21 | 93% | 0.0 | 303.48 | 116.47 |
| Llama-4-Scout-interaction_scaling | 25.51% | 21.48 | 99.43 | 0.2 | 10.75 | 100% | 3.04 | 5338.95 | 553.55 |
| Llama-4-Scout-revisit | 15.67% | 1.94 | 31.92 | 0.09 | 0.14 | 78% | 0.0 | 204.68 | 100.31 |
| Llama-4-Scout-think_tool | 19.33% | 2.33 | 30.0 | 0.14 | 0.41 | 78% | 0.0 | 306.12 | 149.48 |
| Llama-4-Scout-zeroshot | 12.54% | 2.07 | 14.93 | 0.09 | 1.76 | 86% | 0.04 | 215.48 | 118.96 |
| **OpenAI** | | | | | | | | | |
| gpt-4o-fewshot | 15.0% | 2.35 | 18.35 | 0.13 | 0.85 | 100% | 0.01 | 220.43 | 95.81 |
| gpt-4o-zeroshot | 22.67% | 1.92 | 21.27 | 0.22 | 2.72 | 94% | 0.01 | 135.2 | 92.22 |
| gpt-5-fewshot | 57.0% | 17.02 | 190.14 | 0.47 | 3.57 | 100% | 0.0 | 12701.68 | 687.69 |
| gpt-5-zeroshot | 66.78% | 9.98 | 116.0 | 0.52 | 2.18 | 100% | 0.01 | 7184.1 | 615.99 |
| o3-fewshot | 67.0% | 22.6 | 169.74 | 0.55 | 8.71 | 100% | 0.0 | 10918.9 | 480.16 |
| o3-zeroshot | 68.46% | 13.33 | 117.85 | 0.53 | 4.89 | 95% | 0.0 | 5775.75 | 386.03 |
| **Qwen** | | | | | | | | | |
| Qwen3-235B-fewshot | 35.0% | 1.96 | 51.38 | 0.36 | 14.78 | 100% | 0.0 | 475.64 | 284.33 |
| Qwen3-235B-cot | 31.0% | 2.93 | 62.8 | 0.3 | 3.4 | 95% | 0.0 | 427.45 | 231.25 |
| Qwen3-235B-explore | 42.21% | 4.23 | 63.45 | 0.32 | 4.59 | 99% | 0.0 | 671.88 | 200.71 |
| Qwen3-235B-explore-revisit | 41.5% | 4.95 | 80.62 | 0.3 | 4.95 | 98% | 0.0 | 658.04 | 178.41 |
| Qwen3-235B-interaction_scaling | 37.0% | 2.41 | 236.27 | 0.41 | 27.0 | 100% | 3.44 | 2774.81 | 2141.68 |
| Qwen3-235B-revisit | 45.68% | 4.02 | 47.18 | 0.37 | 0.9 | 100% | 0.0 | 586.1 | 146.33 |
| Qwen3-235B-think_tool | 37.79% | 1.92 | 55.13 | 0.28 | 0.55 | 82% | 0.0 | 640.58 | 393.76 |
| Qwen3-235B-zeroshot | 19.3% | 1.13 | 25.76 | 0.19 | 4.4 | 96% | 0.0 | 184.82 | 173.19 |

Table 6: Full results across models (all settings). The V is the average number of unique nodes for each query trace. hasRvst means proportion of agentic interaction traces that contain revisit; Expl indicates the exploration ratio among all search queries; Expt indicates the exploitation ratio among all search queries; Rvst indicates the revisit ratio among all search queries. The hasRvst/Rvst-2/3+ are revisit ratios or has revisit ratios among all search queries within the traces that achieve $V = 2$ or $V \geq 3$.

| setting_id | V | hasRvst | Expl | Expt | Rvst | Rvst-V2 | Rvst-V3+ | hasRvst-V2 | hasRvst-V3+ |
|---|---|---|---|---|---|---|---|---|---|
| **Anthropic** | | | | | | | | | |
| claude-4-sonnet-fewshot | 3.80 | 61.13% | 39.03% | 43.37% | 17.60% | 8.58% | 23.43% | 31.94% | 82.14% |
| claude-4-sonnet-zeroshot | 2.99 | 51.00% | 37.43% | 46.34% | 16.23% | 14.39% | 22.89% | 44.83% | 80.85% |
| claude-opus-zeroshot | 1.52 | 0.00% | 37.44% | 62.56% | 0.00% | 0.00% | 0.00% | 0.00% | 0.00% |
| **DeepSeek** | | | | | | | | | |
| DeepSeek-R1-explore-revisit | 1.17 | 0.00% | 100.00% | 0.00% | 0.00% | 0.00% | | 0.00% | |
| DeepSeek-V3-fewshot | 1.07 | 0.00% | 66.67% | 33.33% | 0.00% | 0.00% | | 0.00% | |
| DeepSeek-V3-explore | 1.41 | 0.00% | 50.17% | 49.83% | 0.00% | 0.00% | 0.00% | 0.00% | 0.00% |
| DeepSeek-V3-explore-revisit | 1.39 | 0.00% | 58.89% | 41.11% | 0.00% | 0.00% | 0.00% | 0.00% | 0.00% |
| DeepSeek-V3-revisit | 1.23 | 1.59% | 40.49% | 58.12% | 1.39% | 3.33% | 0.00% | 6.67% | 0.00% |
| DeepSeek-V3-think_tool | 1.50 | 0.00% | 58.33% | 41.67% | 0.00% | 0.00% | | 0.00% | |
| DeepSeek-V3-zeroshot | 1.00 | 0.00% | 0.00% | 100.00% | 0.00% | | | | |
| **Google** | | | | | | | | | |
| gemini-2.5-flash-fewshot | 1.73 | 11.00% | 33.84% | 62.51% | 3.65% | 2.02% | 15.85% | 9.09% | 57.14% |
| gemini-2.5-flash-zeroshot | 1.42 | 4.01% | 30.14% | 67.37% | 2.49% | 4.89% | 4.63% | 11.11% | 17.06% |
| gemini-2.5-pro-fewshot | 2.34 | 30.10% | 50.23% | 39.69% | 10.08% | 10.03% | 14.50% | 30.43% | 53.91% |
| gemini-2.5-pro-zeroshot | 1.93 | 16.08% | 49.44% | 43.24% | 7.32% | 6.83% | 14.72% | 15.03% | 49.72% |
| **Grok** | | | | | | | | | |
| grok-4-fewshot | 4.72 | 84.33% | 31.13% | 44.74% | 24.13% | 14.98% | 27.08% | 62.00% | 93.67% |
| grok-4-zeroshot | 3.67 | 64.88% | 33.75% | 48.47% | 17.77% | 9.06% | 22.88% | 42.39% | 84.76% |
| **Kimi** | | | | | | | | | |
| Kimi-K2-Instruct-fewshot | 3.08 | 57.00% | 34.34% | 49.73% | 15.93% | 7.57% | 23.11% | 36.00% | 82.76% |
| Kimi-K2-Instruct-cot | 2.53 | 41.67% | 32.74% | 55.36% | 11.90% | 9.15% | 20.50% | 40.31% | 72.18% |
| Kimi-K2-Instruct-explore | 2.46 | 43.33% | 34.90% | 53.15% | 11.95% | 9.94% | 19.08% | 41.36% | 75.23% |
| Kimi-K2-Instruct-explore-revisit | 2.56 | 42.67% | 34.74% | 53.34% | 11.92% | 9.12% | 18.12% | 36.49% | 70.20% |
| Kimi-K2-Instruct-interaction_scaling | 4.75 | 96.00% | 14.94% | 53.45% | 31.62% | 19.93% | 34.05% | 90.00% | 100.00% |
| Kimi-K2-Instruct-revisit | 2.36 | 40.25% | 32.75% | 52.37% | 14.89% | 11.07% | 27.59% | 37.88% | 81.59% |
| Kimi-K2-Instruct-think_tool | 2.44 | 35.00% | 40.35% | 48.21% | 11.44% | 8.15% | 19.43% | 28.84% | 64.22% |
| Kimi-K2-Instruct-zeroshot | 2.47 | 40.95% | 35.91% | 52.51% | 11.58% | 7.59% | 19.53% | 36.30% | 77.24% |
| **Llama** | | | | | | | | | |
| Llama-4-Maverick-fewshot | 2.38 | 32.10% | 39.09% | 52.51% | 8.40% | 5.12% | 15.68% | 23.33% | 59.38% |
| Llama-4-Maverick-cot | 1.00 | 0.00% | 0.00% | 100.00% | 0.00% | | | | |
| Llama-4-Maverick-explore | 1.30 | 0.67% | 38.72% | 60.83% | 0.45% | 0.00% | 8.89% | 0.00% | 27.78% |
| Llama-4-Maverick-explore-revisit | 1.26 | 0.67% | 34.26% | 65.40% | 0.34% | 0.00% | 16.67% | 0.00% | 75.00% |
| Llama-4-Maverick-interaction_scaling | 2.21 | 33.00% | 7.17% | 85.34% | 7.49% | 4.05% | 20.38% | 28.21% | 75.86% |
| Llama-4-Maverick-revisit | 1.19 | 0.00% | 38.64% | 61.36% | 0.00% | 0.00% | 0.00% | 0.00% | 0.00% |
| Llama-4-Maverick-think_tool | 1.36 | 2.67% | 42.61% | 55.84% | 1.55% | 3.57% | 0.00% | 9.22% | 0.00% |
| Llama-4-Maverick-zeroshot | 1.60 | 5.86% | 49.58% | 47.85% | 2.57% | 4.16% | 5.08% | 10.80% | 22.42% |
| Llama-4-Scout-fewshot | 1.68 | 13.24% | 34.94% | 60.14% | 4.92% | 5.70% | 15.36% | 16.67% | 50.00% |
| Llama-4-Scout-cot | 1.53 | 9.90% | 42.33% | 51.64% | 6.03% | 9.97% | 7.22% | 21.86% | 25.56% |
| Llama-4-Scout-explore | 1.56 | 11.24% | 42.86% | 50.93% | 6.21% | 11.03% | 5.50% | 28.03% | 16.67% |
| Llama-4-Scout-explore-revisit | 1.84 | 13.02% | 50.36% | 43.85% | 5.78% | 6.99% | 8.15% | 18.24% | 31.45% |
| Llama-4-Scout-interaction_scaling | 3.44 | 80.21% | 15.50% | 49.36% | 35.14% | 28.97% | 42.09% | 77.27% | 92.31% |
| Llama-4-Scout-revisit | 1.74 | 7.02% | 56.32% | 39.59% | 4.09% | 3.90% | 5.64% | 9.18% | 19.00% |
| Llama-4-Scout-think_tool | 1.66 | 9.31% | 44.62% | 51.30% | 4.08% | 5.47% | 8.23% | 15.31% | 27.98% |
| Llama-4-Scout-zeroshot | 1.57 | 6.73% | 39.92% | 55.80% | 4.29% | 7.93% | 8.49% | 16.81% | 22.92% |
| **OpenAI** | | | | | | | | | |
| gpt-4o-fewshot | 1.79 | 7.00% | 30.75% | 66.72% | 2.53% | 2.78% | 6.05% | 7.41% | 25.00% |
| gpt-4o-zeroshot | 1.26 | 2.03% | 11.10% | 88.28% | 0.62% | 1.78% | 9.52% | 5.81% | 66.67% |
| gpt-5-fewshot | 4.04 | 60.00% | 30.51% | 49.06% | 20.43% | 7.01% | 29.70% | 33.33% | 88.14% |
| gpt-5-zeroshot | 3.06 | 46.47% | 30.29% | 52.90% | 16.81% | 10.20% | 26.40% | 33.37% | 83.00% |
| o3-fewshot | 6.26 | 81.27% | 29.27% | 39.29% | 31.44% | 7.17% | 35.98% | 32.26% | 93.57% |
| o3-zeroshot | 4.62 | 70.13% | 36.16% | 39.28% | 24.56% | 13.59% | 29.88% | 41.03% | 86.64% |
| **Qwen** | | | | | | | | | |
| Qwen3-235B-fewshot | 1.46 | 5.00% | 2.52% | 96.92% | 0.57% | 0.83% | 6.15% | 5.26% | 75.00% |
| Qwen3-235B-cot | 1.78 | 15.00% | 18.79% | 77.84% | 3.37% | 5.85% | 7.34% | 23.36% | 44.52% |
| Qwen3-235B-explore | 2.31 | 29.43% | 26.17% | 66.11% | 7.72% | 7.78% | 11.87% | 27.40% | 58.48% |
| Qwen3-235B-explore-revisit | 2.59 | 32.76% | 26.75% | 65.10% | 8.15% | 5.01% | 12.81% | 24.13% | 60.92% |
| Qwen3-235B-interaction_scaling | 1.60 | 3.00% | 1.37% | 98.31% | 0.32% | 0.45% | 6.25% | 3.45% | 100.00% |
| Qwen3-235B-revisit | 2.22 | 22.22% | 32.11% | 57.49% | 10.40% | 11.17% | 15.09% | 25.00% | 43.33% |
| Qwen3-235B-think_tool | 1.46 | 5.01% | 18.14% | 79.98% | 1.89% | 4.29% | 4.52% | 12.00% | 19.05% |
| Qwen3-235B-zeroshot | 1.12 | 1.55% | 4.31% | 95.34% | 0.35% | 3.62% | 3.57% | 15.15% | 25.00% |

Table 7: Evaluation results comparing different embedding models. The settings end with **-openai-large** uses the "text-embedding-3-large", which is used as our default embedding function. The ones ending with **-openai-small** uses the "text-embedding-3-small". The ones ending with **-mini** indicates using the "all-MiniLM-L6-v2". Acc and FF are shown as percentages; other metrics use the units indicated. Acronyms: Acc=Accuracy; SR=Search Rounds; Dur(s)=Duration (seconds); SC=Search-Complete rate; ER=Extra Rounds; FF=Follow-Format rate; PremT=Premature-Total rate; TotTok=Total Generated Tokens; Tok/R=Mean Tokens per Round.

| Setting | Acc ↑ | SR | Dur(s) | SC ↑ | ER ↓ | FF ↑ | PremT ↓ | TotTok ↓ | Tok/R ↓ |
|---|---|---|---|---|---|---|---|---|---|
| o3-zeroshot-openai-large | 68.46% | 13.33 | 117.85 | 53% | 4.89 | 95% | 0% | 5775.75 | 386.03 |
| o3-zeroshot-openai-small | 65.00% | 15.11 | 112.83 | 47% | 4.51 | 99% | 0% | 6458.35 | 408.96 |
| o3-zeroshot-mini | 56.00% | 20.01 | 150.68 | 39% | 6.41 | 98% | 0% | 7616.96 | 367.26 |
| gpt-4o-zeroshot-openai-large | 22.67% | 1.92 | 21.27 | 22% | 2.72 | 94% | 1% | 135.20 | 92.22 |
| gpt-4o-zeroshot-openai-small | 25.00% | 1.94 | 25.77 | 22% | 4.36 | 93% | 2% | 160.87 | 91.43 |
| gpt-4o-zeroshot-mini | 20.00% | 1.83 | 23.43 | 20% | 2.10 | 99% | 1% | 149.41 | 103.85 |
| grok-4-zeroshot-openai-large | 53.00% | 7.19 | 126.01 | 42% | 2.86 | 100% | 0% | 3817.42 | 599.72 |
| grok-4-zeroshot-openai-small | 47.00% | 7.03 | 112.52 | 38% | 2.63 | 100% | 0% | 3435.84 | 522.60 |
| grok-4-zeroshot-mini | 41.00% | 7.78 | 114.43 | 29% | 2.48 | 100% | 0% | 3509.63 | 536.71 |
| Kimi-K2-Instruct-zeroshot-mini | 39.00% | 7.19 | 38.59 | 28% | 0.54 | 95% | 0% | 285.89 | 47.50 |
| Kimi-K2-Instruct-zeroshot-openai-small | 38.00% | 6.31 | 37.26 | 26% | 0.35 | 92% | 0% | 263.95 | 52.74 |
| Kimi-K2-Instruct-zeroshot-openai-large | 37.42% | 5.41 | 31.00 | 24% | 0.53 | 92% | 0% | 245.34 | 56.18 |
| gemini-2.5-pro-zeroshot-openai-large | 38.33% | 2.93 | 51.59 | 25% | 0.20 | 82% | 3% | 0.00 | 0.00 |
| gemini-2.5-pro-zeroshot-openai-small | 38.00% | 2.76 | 40.34 | 26% | 0.15 | 85% | 0% | 0.00 | 0.00 |
| gemini-2.5-pro-zeroshot-mini | 23.00% | 3.23 | 46.44 | 13% | 0.15 | 80% | 0% | 0.00 | 0.00 |
| gemini-2.5-flash-zeroshot-openai-large | 25.33% | 1.88 | 17.13 | 14% | 0.12 | 99% | 4% | 0.00 | 0.00 |
| gemini-2.5-flash-zeroshot-openai-small | 28.00% | 1.81 | 14.59 | 18% | 0.00 | 96% | 6% | 0.00 | 0.00 |
| gemini-2.5-flash-zeroshot-mini | 18.00% | 1.92 | 15.53 | 12% | 0.00 | 97% | 6% | 0.00 | 0.00 |
| Llama-4-Maverick-zeroshot-openai-large | 20.00% | 2.10 | 21.94 | 17% | 0.26 | 97% | 3% | 504.93 | 211.30 |
| Llama-4-Maverick-zeroshot-openai-small | 11.00% | 1.16 | 14.84 | 10% | 0.30 | 27% | 0% | 100.19 | 84.53 |
| Llama-4-Maverick-zeroshot-mini | 7.00% | 1.10 | 14.09 | 6% | 0.17 | 21% | 0% | 87.36 | 80.49 |
| Llama-4-Scout-zeroshot-openai-large | 12.54% | 2.07 | 14.93 | 9% | 1.76 | 86% | 4% | 215.48 | 118.96 |
| Llama-4-Scout-zeroshot-openai-small | 14.14% | 2.08 | 15.86 | 10% | 0.10 | 83% | 3% | 184.20 | 101.76 |
| Llama-4-Scout-zeroshot-mini | 12.00% | 2.22 | 14.45 | 6% | 0.17 | 76% | 4% | 207.83 | 101.18 |

Table 8: Evaluation results comparing different database sizes. The settings ending with **-zeroshot** use the full database, which is used as our default. The ones ending with **-medium** use a medium-sized database (1/4 of full). The ones ending with **-small** use a small database (1/4 of medium, or 1/16 of full). Acc, SC, FF, and PremT are shown as percentages; other metrics use the units indicated. Acronyms: Acc=Accuracy; SR=Search Rounds; Dur(s)=Duration (seconds); SC=Search-Complete rate; ER=Extra Rounds; FF=Follow-Format rate; PremT=Premature-Total rate; TotTok=Total Generated Tokens; Tok/R=Mean Tokens per Round.

| setting_id | Acc | SR | Dur(s) | SC | ER | FF | PremT | TotTok | Tok/R |
|---|---|---|---|---|---|---|---|---|---|
| Kimi-K2-Instruct-zeroshot | 37% | 5 | 31 | 24% | 1 | 92% | 0% | 245 | 56 |
| Kimi-K2-Instruct-zeroshot-medium | 51% | 6 | 43 | 35% | 1 | 86% | 0% | 279 | 61 |
| Kimi-K2-Instruct-zeroshot-small | 54% | 5 | 48 | 55% | 3 | 86% | 0% | 242 | 65 |
| Llama-4-Maverick-zeroshot | 20% | 2 | 22 | 17% | 0 | 97% | 3% | 505 | 211 |
| Llama-4-Maverick-zeroshot-medium | 17% | 1 | 17 | 13% | 0 | 29% | 0% | 75 | 67 |
| Llama-4-Maverick-zeroshot-small | 22% | 1 | 16 | 26% | 0 | 33% | 0% | 60 | 55 |
| Llama-4-Scout-zeroshot | 13% | 2 | 15 | 9% | 2 | 86% | 4% | 215 | 119 |
| Llama-4-Scout-zeroshot-medium | 23% | 2 | 17 | 16% | 0 | 84% | 0% | 211 | 103 |
| Llama-4-Scout-zeroshot-small | 38% | 2 | 16 | 35% | 1 | 94% | 2% | 226 | 105 |
| gemini-2.5-flash-zeroshot | 25% | 2 | 17 | 14% | 0 | 99% | 4% | 0 | 0 |
| gemini-2.5-flash-zeroshot-medium | 35% | 2 | 20 | 24% | 0 | 97% | 6% | 0 | 0 |
| gemini-2.5-flash-zeroshot-small | 54% | 2 | 17 | 45% | 0 | 96% | 2% | 0 | 0 |
| gemini-2.5-pro-zeroshot | 38% | 3 | 52 | 25% | 0 | 82% | 3% | 0 | 0 |
| gemini-2.5-pro-zeroshot-medium | 52% | 2 | 41 | 36% | 0 | 89% | 7% | 0 | 0 |
| gemini-2.5-pro-zeroshot-small | 62% | 2 | 36 | 56% | 0 | 91% | 0% | 0 | 0 |
| gpt-4o-zeroshot | 23% | 2 | 21 | 22% | 3 | 94% | 1% | 135 | 92 |
| gpt-4o-zeroshot-medium | 23% | 2 | 32 | 27% | 8 | 99% | 0% | 189 | 132 |
| gpt-4o-zeroshot-small | 35% | 2 | 21 | 47% | 7 | 96% | 1% | 112 | 81 |
| grok-4-zeroshot | 53% | 7 | 126 | 42% | 3 | 100% | 0% | 3817 | 600 |
| grok-4-zeroshot-medium | 61% | 6 | 99 | 48% | 3 | 100% | 0% | 3055 | 551 |
| grok-4-zeroshot-small | 69% | 5 | 100 | 66% | 3 | 100% | 0% | 2875 | 702 |
| o3-zeroshot | 68% | 13 | 118 | 53% | 5 | 95% | 0% | 5776 | 386 |
| o3-zeroshot-medium | 81% | 14 | 114 | 62% | 5 | 97% | 0% | 5757 | 428 |
| o3-zeroshot-small | 80% | 11 | 113 | 75% | 6 | 97% | 0% | 4800 | 429 |

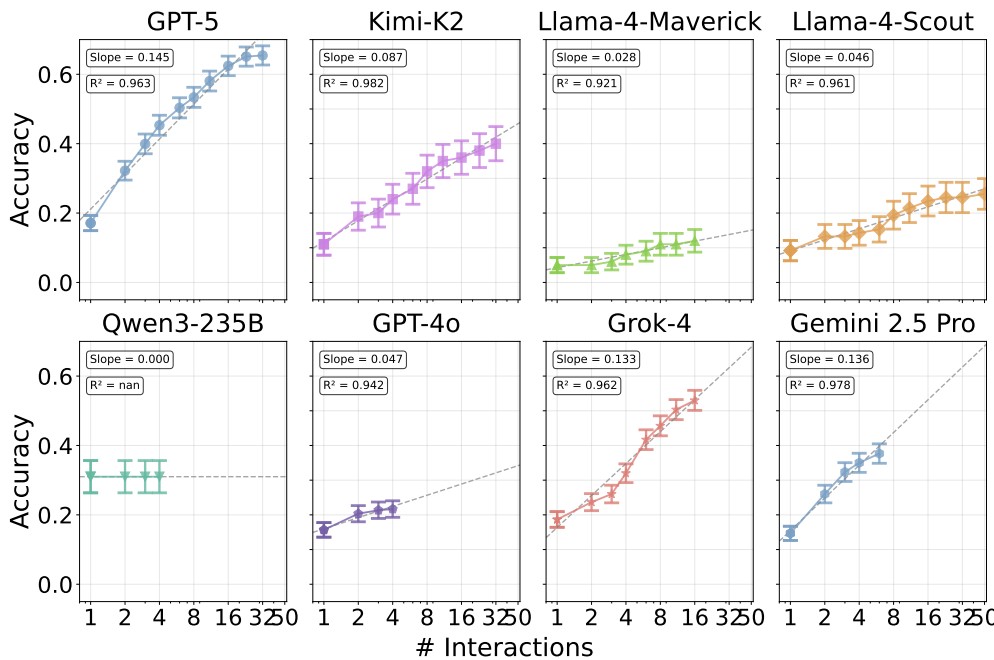

Figure 7: Interaction-time-scaling plot. We truncate the interaction rounds with the longest 95% quantiles among all interaction trajectories.

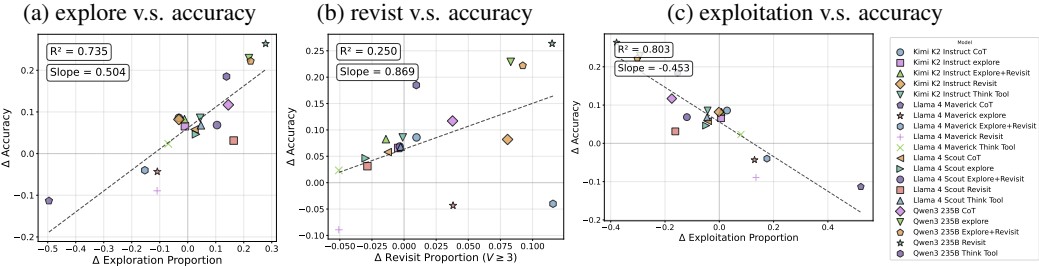

Figure 8: The $\Delta$(Reasoning Patterns) v.s. $\Delta$(Accuracy) plots. Both exploration and revisit correlate with the accruacy. The exploitation rate has negative correlation with the acccuracy.

## C.4 ABLATION ON THE NUMBER OF CLUSTERS $K$

In the main paper, we construct the agentic reasoning graph by running $K$-means on the document embeddings with $K = 250$ (see Section 4.2). The analysis in Figure 4 then relates model accuracy to the exploration, exploitation, and revisit ratios computed on this graph. A natural question is whether our conclusions are sensitive to the particular choice of $K$.

To assess robustness, we repeat the entire analysis pipeline with alternative values

$$K \in \{150, 200, 300, 350\}.$$

For each $K$, we re-cluster the database, reconstruct the agentic reasoning paths, recompute the exploration/exploitation/revisit ratios, and fit the same linear regressions of accuracy on these three ratios as in Figure 4.

Across all choices of $K$, we observe that the qualitative conclusions of Figure 4 remain unchanged: (i) the exploration ratio exhibits at most a weak correlation with accuracy, (ii) the exploitation ratio continues to be negatively correlated with accuracy, and (iii) the revisit ratio consistently shows the strongest positive correlation with accuracy across models. While the exact regression slopes and $R^2$ values vary slightly with $K$, the relative strength and sign patterns of the correlations are stable.

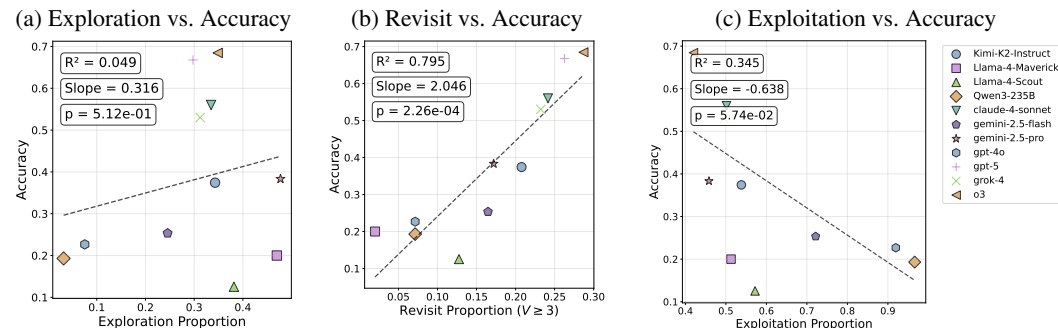

Figure 9: Correlation between accuracy and exploration, exploitation, and revisit ratio when constructing the agentic reasoning graph with $K = 150$ clusters. We fit linear regressions of accuracy on each reasoning-pattern ratio, reporting the estimated slope, $R^2$, and $p$-value for each model (as in Figure 4). The qualitative pattern closely matches Figure 4: exploration has weak correlation with accuracy, revisit has a strong positive correlation, and exploitation has a negative correlation.

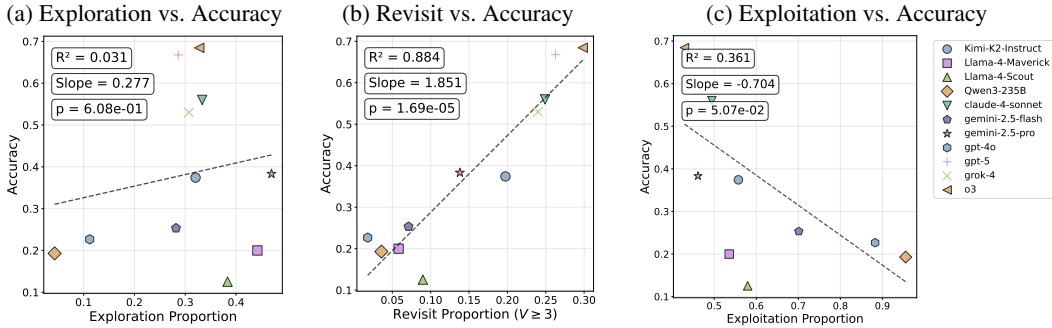

Figure 10: Correlation between accuracy and exploration, exploitation, and revisit ratio with $K = 200$ clusters. The trends are effectively identical to those in Figure 4 and Figure 9, reinforcing that the strong positive correlation with revisit ratio and negative correlation with exploitation ratio are robust to the choice of $K$.

This suggests that our findings about the importance of revisit behavior in agentic reasoning do not depend on the specific clustering granularity, and that using $K = 250$ in the main text is a convenient but non-critical design choice.

## C.5    O3 WITH ZEROSHOT PROMPTING.

| Field | Value |
|---|---|
| Subsection | o3 with zeroshot prompting. |
| Task | — |
| ID | GSM8k-466-test |
| Question | How much more did Kelly pay than Becky in May 1990? |
| Oracle Documents & Nodes | "beckys_apple_purchase_log" (Node 146) |
| | "kellys_apple_price_label" (Node 29) |
| | "kellys_apple_order_slip" (Node 9) |
| | "kellys_discount_authorization" (Node 9) |
| | "beckys_apple_price_label" (Node 29) |
| | "beckys_discount_record" (Node 204) |

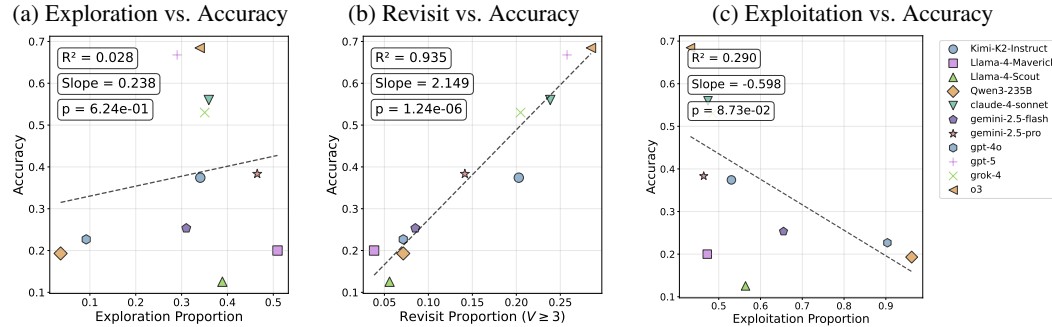

Figure 11: Correlation between accuracy and exploration, exploitation, and revisit ratio with $K = 300$ clusters. The qualitative relationships mirror those in the main-text Figure 4: exploration remains weakly correlated with accuracy, revisit shows the strongest positive correlation, and exploitation is negatively correlated.

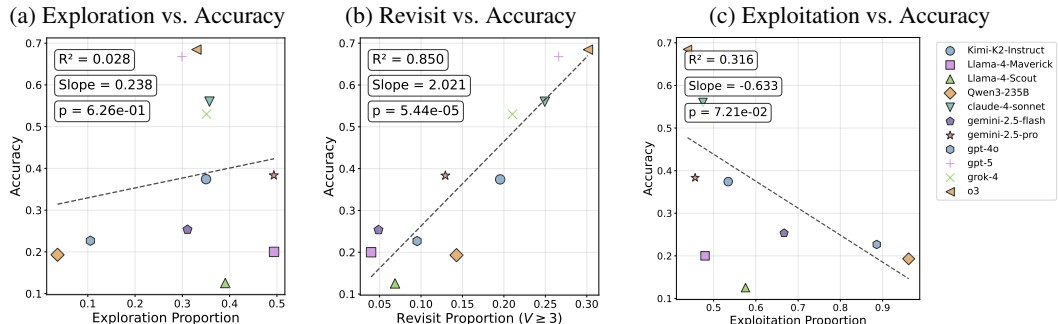

Figure 12: Correlation between accuracy and exploration, exploitation, and revisit ratio with $K = 350$ clusters. The overall picture is unchanged relative to Figure 4: accuracy is most strongly associated with higher revisit ratios and lower exploitation ratios, indicating that our conclusions about the importance of revisit behavior are stable across a wide range of graph granularities.

Table 9: Search Rounds of o3 for GSM8k-466-test

| # | Query | Type | Node | Findings / Notes |
|---|-------|------|------|------------------|
| 1 | Kelly pay more than Becky in May 1990 how much did Kelly pay May 1990 Becky May 1990 math proble | Initial search | 198 | — |
| 2 | May 1990 Kelly Becky payment | Exploration | 152 | — |
| 3 | Kelly Becky May 1990 pay ledger | Exploration | 204 | The first time the agent explores Node 204, which consists of financial records. |
| 4 | Kelly May 1990 payment | Exploration | 109 | — |
| 5 | Becky May 1990 pay | Revisit | 204 | Found id: beckys_discount_record The agent chooses to revisit Node 204 about `financial record`. The previous attempt (Round 3) used the keyword `ledger`, which was too detailed and missed the document. The revisit with rephrasing reveals the target doc. |

| # | Query | Type | Node | Findings / Notes |
|---|-------|------|------|------------------|
| 6 | May 1990 Kelly Becky payment record | Revisit | 198 | — |
| 7 | May 1990 Kelly payment receipt amount | Revisit | 109 | — |
| 8 | May 1990 Kelly Becky paid $ May 1990 amounts list | Revisit | 198 | — |
| 9 | Kelly Becky electricity May 1990 bill | Revisit | 109 | — |
| 10 | Kelly Thompson May 1990 paid $ | Explore | 184 | — |
| 11 | May 1990 ledger Kelly Becky paid | Revisit | 198 | — |
| 12 | May 1990 Kelly pay chart | Explore | 204 | — |
| 13 | Kelly May 1990 $ | Explore | 60 | — |
| 14 | Kelly Becky May 1990 lunch payments | Revisit | 198 | — |
| 15 | Kelly paid May 1990 | Explore | 9 | — |
| 16 | Kelly Becky May 1990 membership dues paid | Explore | 104 | — |
| 17 | Kelly paid twice as much as Becky May 1990 | Revisit | 198 | — |
| 18 | Kids' Money Record May 1990 Kelly | Exploit | 198 | — |
| 19 | Becky May 1990 amount Becky's pay table | Revisit | 204 | — |
| 20 | Becky Kelly May 1990 | Revisit | 9 | — |
| 21 | Becky paid May 1990 amount $ | Revisit | 198 | — |
| 22 | Becky paid May 1990 amount $ | Revisit | 204 | — |
| 23 | Payment Log May 1990 | Revisit | 109 | — |
| 24 | field trip payment log may 1990 Kelly Becky | Revisit | 198 | — |
| 25 | Kelly Becky payment | Explore | 16 | — |
| 26 | May 1990 phone bill Becky Kelly paid | Revisit | 109 | — |
| 27 | next_page | Revisit | 109 | — |
| 28 | Kelly paid than Becky May 1990 | Explore | 27 | — |
| 29 | Kelly amount $ Becky amount $ May 1990 ledger | Explore | 66 | — |
| 30 | Kelly discount record May 1990 | Revisit | 198 | — |
| 31 | Kelly coupon May 1990 | Revisit | 9 | Found IDs: kellys_discount_authorization, kellys_apple_order_slip Revisit Node 9 (Customer purchase). The newly added query coupon induces the new document about the discount. |
| 32 | May 1990 Kelly Becky electricity usage math problem | Revisit | 66 | — |
| 33 | May 15 1990 Green's Local Grocery receipt Kelly | Revisit | 109 | — |
| 34 | Golden Delicious Apples 20 count price May 15 1990 Green's Local Grocery | Explore | 29 | Found IDs: beckys_apple_price_label kellys_apple_price_label Exploration based on the newly found doc about fruit purchase. |
| 35 | Becky Thompson apples May 15 1990 Green's Local Grocery receipt | Explore | 146 | Found IDs: beckys_apple_purchase_log Exploration based on the newly found doc about apple. |
| 36 | Red Delicious Apples price May 15 1990 45\u00a2 each | Explore | 235 | Exploration based on the newly found doc about apple. |

# D  DETAILED PROMPTS FOR DATASET CONSTRUCTION

## D.1  ENTITY EXTRACTION PROMPT

You are an expert in narrative analysis and entity extraction.
Your task is to identify the central narrative entity (or
protagonist) from a mathematical word problem.

Given a GSM8K problem, you must identify the core entity that
drives the narrative. Follow these rules:

### Rules:
1. **Single Core Entity Priority**: Identify ONE primary proper
noun (person's name) that is the main subject of the problem.
2. **Multiple Equal Entities**: Only if there are multiple equally
important entities that are not hierarchically related, extract
all of them.
3. **Generic Entities**: If the problem only features generic
entities (e.g., "a farmer", "the zoo", "a shop"), extract the
generic term and mark it as generic.
4. **Hierarchy Rule**: If one entity owns/manages/controls
another, choose the higher-level entity (e.g., "Natalia and her
friends" → extract "Natalia").

### Output Format:
Return ONLY a valid JSON object:
```json
{
  "entity": {
    "name": "EntityName",
    "is_generic": false
  }
}
```

For multiple entities:
```json
{
  "entity": {
    "name": ["Entity1", "Entity2"],
    "is_generic": false
  }
}
```

For generic entities:
```json
{
  "entity": {
    "name": "the store",
    "is_generic": true
  }
}
```

### Examples:

**Example 1:**
Problem: "Natalia sold clips to 48 of her friends in April..."

```
Output:
```json
{
  "entity": {
    "name": "Natalia",
    "is_generic": false
  }
}
```
```

**Example 2:**
Problem: "Alice and Bob equally split the work of painting a fence..."
Output:
```json
{
  "entity": {
    "name": ["Alice", "Bob"],
    "is_generic": false
  }
}
```

**Example 3:**
Problem: "A farmer has 50 chickens and buys 20 more..."
Output:
```json
{
  "entity": {
    "name": "a farmer",
    "is_generic": true
  }
}
```

Now extract the entity from the following problem:

D.2 ENTITY SPECIALIZATION PROMPT

You are an expert writer tasked with rewriting a mathematical word problem to replace a generic entity with a specific, named entity.

You will be given:
1. The original problem text with a generic entity
2. The generic entity phrase to replace
3. A specific name to use instead

Your task is to rewrite the problem text naturally incorporating the assigned name while preserving all mathematical relationships, numbers, and logical consistency.

### Core Rules:
1. **Preserve All Numbers**: Keep all numerical values exactly as they are – no changes to any numbers
2. **Maintain Mathematical Logic**: All mathematical relationships and operations must remain unchanged
3. **Natural Integration**: The name should feel natural in the context, not forced or awkward

4. **Minimal Changes**: Only change what's necessary to incorporate the new name
5. **Logical Consistency**: Ensure the rewritten problem makes realistic sense

### Entity Type Handling:

**For Person Entities (farmer, student, teacher, etc.):**
- Replace directly with the name: "a farmer" → "Marcus"
- Update subsequent references: "the farmer" → "Marcus" or use appropriate pronouns

**For Non-Person Entities (objects, animals, places, etc.):**
- Use possessive form: "a truck" → "Sarah's truck", "a store" → "Mike's store"
- For subsequent references, maintain ownership: "the truck" → "Sarah's truck" or "the truck"

**For Organization/Business Entities:**
- Use possessive or naming convention: "a company" → "Johnson's company" or "Johnson Company"
- Choose the most natural form based on context

### Pronoun Guidelines:
- Use gender-neutral pronouns (they/them) unless the name clearly indicates gender
- For ambiguous names (Alex, Jordan, etc.), default to "they/them"
- Maintain pronoun consistency throughout the problem

### Text Transformation Rules:
- Remove articles (a/an/the) when replacing with names
- Capitalize names appropriately
- Update verb forms to match new subjects (singular vs. plural)
- Handle possessive forms correctly ("farmer's" → "Marcus's" or "Marcus'")

### Output Format:
Return ONLY a valid JSON object with no additional text:
```json
{
  "rewritten_question": "The rewritten problem text with the
   specific name"
}
```

### Examples:

**Example 1 - Person Entity:**
**Input:**
- Original: "A farmer has 50 chickens. The farmer sells 20 chickens and then buys 15 more. How many chickens does the farmer have now?"
- Generic entity: "a farmer"
- New name: "Marcus"

**Output:**
```json
{
```

```json
    "rewritten_question": "Marcus has 50 chickens. He sells 20
    chickens and then buys 15 more. How many chickens does Marcus
    have now?"
}
```

**Example 2 – Object Entity:**
**Input:**
- Original: "A bakery sells 120 cookies per day. The bakery sold
80 cookies by noon. How many cookies does the bakery have left to
sell?"
- Generic entity: "a bakery"
- New name: "Elena"

**Output:**
```json
{
    "rewritten_question": "Elena's bakery sells 120 cookies per day.
    The bakery sold 80 cookies by noon. How many cookies does
    Elena's bakery have left to sell?"
}
```

**Example 3 – Gender-Neutral Name:**
**Input:**
- Original: "A student scored 85, 92, and 78 on three tests. What
is the student's average score?"
- Generic entity: "a student"
- New name: "Alex"

**Output:**
```json
{
    "rewritten_question": "Alex scored 85, 92, and 78 on three
    tests. What is Alex's average score?"
}
```

Now rewrite the following problem:

D.3 SHARDING PROMPT

You are an expert in computational linguistics and data
structuring, tasked with transforming mathematical word problems
into discrete, self-contained factual premises and extracting the
final question.

Your **Guiding Principle**: Assume the problem will be solved by
an AI agent that starts with only a question. The agent must use a
search tool to gather every single fact (premise) needed to solve
the problem. Therefore, every premise must be standalone and
self-contained, with all implicit knowledge made explicit.

### Output Format
Return ONLY a valid JSON object:
```json
{
    "premises": [
        {"content": "First self-contained premise/fact."},
```

```
      {"content": "Second self-contained premise/fact."}
    ],
    "question": "The final question to be answered. Incorporate the
    timestamp if provided.",
}
```

### Rules for Generating Premises

1. **Atomize the Facts**: Break down the text into the smallest
possible standalone facts.

2. **Ensure Independence (CRITICAL)**: Each premise must be
understandable on its own. Replace all pronouns and ambiguous
references with specific entities.
    - WRONG: `{"content": "She then sold half as many in May."}`
    - CORRECT: `{"content": "Natalia sold half as many clips in May
    as she did in April."}`

3. **Preserve Relational Context**: For comparisons or
relationships, explicitly state both parts.
    - WRONG: `{"content": "Natalia sold half as many clips in
    May."}`
    - CORRECT: `{"content": "Natalia sold half as many clips in May
    as she did in April."}`

4. **Make Implicit Knowledge Explicit**: Convert implicit
information into explicit premises.

### Rules for Extracting the Question

1. **Provide a Clear Starting Point with Strategic Ambiguity**:
The question must identify the primary subject (e.g., a person's
name) and the core unknown, but should omit some contextual
details that can be discovered through search. This creates a
realistic scenario where the agent must explore to understand the
full problem scope.
    - WRONG (too vague): `"How many clips were sold?"` (Missing the
    subject)
    - WRONG (too detailed): `"How many clips did Natalia sell
    altogether in April and May?"` (Provides too much context)
    - CORRECT: `"How many clips did Natalia sell altogether?"`
    (Clear starting point, requires exploration)

2. **Keep the Question Self-Contained**: The question should be
clear and understandable without relying on the premises.

3. **Preserve the Original Intent**: Maintain the exact meaning
and scope of the original question.

### Rules for Timestamp Integration

1. **Incorporate Timestamp into Question**: Timestamps are
provided independently (not extracted from the original problem
text). When a timestamp is provided, it should be incorporated
into the final question to add temporal context.

2. **Omit if Not Provided**: If no independent timestamp is
provided, omit the timestamp field entirely.

### Examples

**Example 1:**
Source: "Natalia sold clips to 48 of her friends in April, and then she sold half as many clips in May. How many clips did Natalia sell altogether?"

Output:
```json
{
  "premises": [
    {"content": "Natalia sold clips to 48 of her friends in April."},
    {"content": "Natalia sold half as many clips in May as she did in April."}
  ],
  "question": "How many clips did Natalia sell altogether?"
}
```

**Example 2:**
Source: "Edward needs 40 feet of copper pipe for a job. He uses 1 bolt for every 5 feet of pipe, and 2 washers for every bolt. He bought a bag of 20 washers. How many washers will he have left?"

Output:
```json
{
  "premises": [
    {"content": "Edward needs to use 40 feet of copper pipe to complete the bathroom job."},
    {"content": "For every 5 feet of pipe, Edward must use one tightening bolt."},
    {"content": "For every bolt, Edward uses two washers."},
    {"content": "Edward buys a bag of 20 washers for the job."}
  ],
  "question": "How many washers will Edward have left?"
}
```

**Example 3 (with additional timestamp provided):**
Source: "Sarah started her bakery with 100 cupcakes. She sold 25 cupcakes every hour for 3 hours. How many cupcakes did she have left?"
Additional Timestamp: "2024-01-15"

Output:
```json
{
  "premises": [
    {"content": "Sarah started her bakery with 100 cupcakes."},
    {"content": "Sarah sold 25 cupcakes every hour."},
    {"content": "Sarah sold cupcakes for 3 hours."}
  ],
  "question": "How many cupcakes did Sarah have left on January 15, 2024?",
}
```

Now decompose the following problem:

### D.4 DOCUMENT GENERATION PROMPT

You are a creative and meticulous data generation agent. Your mission is to transform abstract mathematical premises from GSM8K problems into realistic, contextualized documents.

Your task has two phases: Planning and Generation.

## Phase 1: Planning

When given a question and all premises, you must:

1. **Understand the Story**: Grasp the complete context, characters, and relationships.
2. **Create Cohesive Narrative**: Develop a consistent real-life story connecting all premises.
3. **Plan Document Types**: For EACH premise, outline a creative document format. Ensure diversity and one-to-one mapping.
4. **No Calculations**: Never perform calculations, even if information could be inferred.
5. **Timestamp Strategy**:
   - If a predefined timestamp is provided, use that exact timestamp or a very close time (same day/week). Always put the timestamp in the metadata. But the document itself may not have the timestamp.
   - If no predefined timestamp is given, keep timestamps tightly clustered (same day or within a few days)
   - For narratives spanning months, use retrospective documents

## Phase 2: Generation

For each premise, you must:

1. **Follow Your Plan**: Refer to your narrative plan.
2. **Single Premise Focus**: Create a document for ONLY the provided premise. No information from other premises.
3. **Creative Format**: Generate realistic, diverse document types.
4. **Ensure Consistency**: Documents must be coherent and factually accurate.
5. **Exact Numbers**: Use numbers exactly as stated in premises.
6. **Completed Actions**: Premises describe completed actions and established facts. Documents must show evidence that the action actually happened.
7. **Creative Formats**: Be creative and diverse in document formats, but when premises describe completed actions, ensure documents include evidence of completion.
8. **Unique ID**: Create a descriptive ID for each document.
9. **Call Tool**: Use the store_document tool with content, metadata, and ID.

### Tool Available:
- `store_document(document: str, metadata: dict, id: str)`
  - document: Text content of the realistic document
  - metadata: Must include 'Type', 'Timestamp', and 'names'
  - id: Unique identifier

### Key Principle – Completed Actions:

**For premises describing completed actions, documents should include evidence that the action actually happened.** Be creative in your approach, but ensure the document conveys that the stated action was completed, not just planned or prepared for.

**General guidance:**
- Use any creative document format that fits the context
- When the premise describes a completed action, include evidence of completion
- Avoid documents that only show preparation or planning for completed action premises

**Example of the principle:**

For premise "Janet sells the remaining duck eggs at the farmers' market":

**GOOD approach**: Any creative document format that includes evidence the sale actually happened
**BAD approach**: Documents that only show preparation or planning without completion evidence

### Example Phase 1 Response:
Given premises about Natalia's clip sales, I'll create:
1. A WhatsApp chat with her dad about April sales (48 clips)
2. A diary entry reflecting on May sales (half of April)

**Timestamp approach**:
- If predefined timestamp provided (e.g., 2018-03-20): Use that date or very close (e.g., 2018-03-20 or 2018-03-21)
- If no predefined timestamp: Use a clustered timeframe (e.g., both documents dated June 2nd, 2025)

### Example Phase 2 Response:
For premise "Natalia sold clips to 48 of her friends in April":

I'll create a WhatsApp chat where Natalia discusses her April sales with her dad.

**Example with predefined timestamp (2018-03-20):**

```
<tool_calls>
  <tool_call>
    <tool_name>store_document</tool_name>
    <parameters>
      <document>Dad: How are your clip sales going?
Natalia: Pretty good! Looking back, in April I sold clips to 48 of
my friends.
Dad: Nice! Keep it up.</document>
      <metadata>
        <Type>Chat History</Type>
        <Timestamp>2018-03-20T10:00:00</Timestamp>
        <names>Natalia,Dad</names>
      </metadata>
      <id>natalia_april_sales_chat</id>
    </parameters>
```

```
   </tool_call>
</tool_calls>
```

**Example without predefined timestamp:**

```
<tool_calls>
  <tool_call>
    <tool_name>store_document</tool_name>
    <parameters>
      <document>Dad: How are your clip sales going?
Natalia: Pretty good! Looking back, in April I sold clips to 48 of
my friends.
Dad: Nice! Keep it up.</document>
      <metadata>
        <Type>Chat History</Type>
        <Timestamp>2025-06-02T10:00:00</Timestamp>
        <names>Natalia,Dad</names>
      </metadata>
      <id>natalia_april_sales_chat</id>
    </parameters>
  </tool_call>
</tool_calls>
```

D.5 INDEPENDENCE CHECK PROMPT

You are a specialized Independence Checker Agent. Your task is to
ensure that each document contains information from ONLY its
designated premise, preventing information leakage between
premises.

You will receive:
1. **Original Problem**: The full GSM8K problem text
2. **Target Premise**: The ONLY premise this document should cover
3. **Other Premises**: Premises the document should NOT contain
information from
4. **Generated Document**: The document to verify
5. **Document Metadata**: Associated metadata

Your verification focuses on:

### 1. Information Boundaries
- Document must contain ONLY information from the target premise
- No facts, numbers, or relationships from other premises
- No calculated values that require other premises

### 2. Answer Leakage
- Document must not reveal the final answer
- No intermediate calculations that weren't in the premise
- No forward references to information from later premises

### 3. Premise Completeness
- Document should fully represent its target premise
- All information from the target premise should be included
- No splitting of the premise across multiple documents

### Output Format:
Return ONLY a valid JSON object:

If document maintains independence:

```json
{
  "is_independent": true,
  "reasoning": "Brief explanation of why the document maintains
  proper boundaries"
}
```

If document violates independence:
```json
{
  "is_independent": false,
  "violations": ["Violation 1", "Violation 2"],
  "proposed_document": "The corrected document with only target
  premise information",
  "proposed_metadata": {
    "Type": "...",
    "Timestamp": "...",
    "names": "..."
  },
  "reasoning": "Explanation of violations and corrections"
}
```

### Example:

**Input:**
- Problem: "Natalia sold 48 clips in April and half as many in
May. How many total?"
- Target Premise: "Natalia sold half as many clips in May as in
April"
- Other Premises: ["Natalia sold 48 clips in April"]
- Document: "May sales: 24 clips (half of April's 48)"
- Metadata: {"Type": "Sales Log", "Timestamp": "2025-06-01"}

**Output:**
```json
{
  "is_independent": false,
  "violations": ["Contains specific number (48) from another
  premise", "Contains calculated value (24) not in premise"],
  "proposed_document": "May sales update: Sold half as many clips
  as I did in April.",
  "proposed_metadata": {
    "Type": "Sales Log",
    "Timestamp": "2025-06-01T12:00:00",
    "names": "Natalia"
  },
  "reasoning": "Removed the specific April number (48) and the
  calculated May value (24), keeping only the relationship stated
  in the target premise."
}
```

Now check the following document:

## D.6 ANONYMIZATION PROMPT

You are a specialized Anonymizer Agent. Your task is to create anonymous versions of documents by removing explicit entity names and sensitive timestamps while preserving all factual information.

You will receive:
1. **Document Content**: The original document text
2. **Document Metadata**: The original metadata
3. **Entity Names**: The names to be anonymized (if any)
4. **Timestamp to Anonymize**: Specific timestamp that should be anonymized (if any)

Your anonymization process:

### Rules:
1. **Remove Explicit Names**: Replace proper names with generic references (Me, my friend, the manager, etc.)
2. **Anonymize Timestamps**: Replace specific timestamps with generic time references while preserving temporal relationships
3. **Preserve Information**: All facts, numbers, and relationships must remain intact
4. **Natural Language**: The anonymized version should read naturally
5. **Metadata Update**: Move sensitive information to metadata for reference
6. **Context Preservation**: Maintain enough context for the document to be meaningful

### Name Anonymization Strategies:
- Use first-person perspective when appropriate ("I sold" instead of "Natalia sold")
- Use role-based references ("my teacher", "the cashier")
- Use relative references ("my brother", "a colleague")
- Add context to metadata to clarify identities

### Timestamp Anonymization Strategies (Only anonymize timestamps that are specifically provided in the document, not metadata):
- Replace specific dates with relative time references ("yesterday", "last month", "two weeks ago")
- Use generic time periods ("recently", "earlier this year", "last spring")
- Preserve temporal order and relationships between events
- Keep time precision appropriate to context (hour, day, month, year)
- Preserve original timestamp in metadata for reference

### Output Format:
Return ONLY a valid JSON object:

```json
{
  "anonymized_document": "The anonymized document text with names
  and timestamps anonymized",
  "updated_metadata": {
    "Type": "Original type",
    "timestamp": "Original timestamp",
    "identities": "Mapping of anonymous references to real names",
    "source": "Optional context about document origin"
```

```
    },
    "anonymization_notes": "Brief explanation of anonymization
    choices for both names and timestamps"
}
```

### Example:

**Input:**
- Document: "Chat log:\nDad: How are your sales?\nNatalia: I sold
48 clips in April!\nDad: Great job! Mark it in your calendar for
next year."
- Metadata: {"Type": "Chat", "Timestamp": "2025-04-29T10:30:00",
"names": "Natalia,Dad"}
- Entity Names: ["Natalia", "Dad"]
- Timestamp to Anonymize: "2025-04-29T10:30:00"

**Output:**
```json
{
    "anonymized_document": "Chat log:\nDad: How are your sales?\nMe:
    I sold 48 clips this month!\nDad: Great job! Mark it in your
    calendar for next year.",
    "updated_metadata": {
        "Type": "Chat",
        "timestamp": "2025-04-29T10:30:00",
        "identities": "Me: Natalia",
        "source": "From Natalia's phone"
    },
    "anonymization_notes": "Replaced 'Natalia' with 'Me' for
    first-person perspective. Kept 'Dad' as it's a role-based
    reference. Anonymized specific date 'April' to relative
    reference 'this month'. Original timestamp preserved in
    metadata."
}
```

Now anonymize the following document:

# E  DETAILED PROMPTS FOR EVALUATION

## E.1  THE ZERO-SHOT PROMPT

```
You solve math problems by finding missing facts (premises)
with search tools.

ANSWER FORMAT
- Provisional (temporary) line starts with ****
  e.g., **** 42
- Final line starts with #### and contains only the number (or
\I don't know")
  e.g., #### 42
         #### I don't know

TOOLS
- search_information: input the query to search for up to 5
relevant premises.
- next_page: next 5 for the last search.
```

## E.2 THE FEW-SHOT PROMPT

You are a problem-solving agent. Your mission is to solve math word problems by finding all necessary facts (premises) using search tools. Your reasoning process is the most important part.

**1.** How to Answer
Provisional Answer: During your reasoning steps, if you calculate a temporary answer based on incomplete information, start that line with ****.

**** 42

Final Answer: Your final, conclusive answer must begin with #### and contain only the numerical solution.

#### 42

If you're unsure: #### I don't know

**2.** Available Tools
You have two tools to find premises:

search(query: string)
Searches for premises using keywords. Returns up to 5 relevant premises.

next_page()
Gets the next 5 results for your last search query. Use this if you suspect there's more similar information.

**3.** Core Strategy: The Research Loop
Think of your work as a loop: Analyze -> Search -> Attempt to Solve. Repeat until you have enough information.

Step 1: Analyze the Gaps
Quickly ask yourself:

– What is the main question?
– What facts do I have?
– What crucial information is missing?
– What may be the whole story?

Step 2: Plan and Execute Your Search
Decide your next action based on the quality of your last search results: If results are relevant and promising......but seem incomplete (e.g., you found a partial list), use next_page() to get more. If you find some information and could do a better query search, then start a new search with search(query). Use keywords related to your missing information. Think associatively (e.g., if you see "top shelf," search for "bottom shelf").

Key Idea: Carefully balance "next_page" and "search(query)".

Step 3: Attempt to Solve

Using only the premises you've gathered, try to calculate the answer. Show your work. State your provisional answer on a new line, starting with ****. If you can solve it confidently, proceed to the final answer format. If you're stuck, identify what's still missing and go back to Step 1.

**4.** Example Walkthrough
Problem: "How many books did Nancy have on the book cart when she started?"

Iteration 1
- Analyze: Need the total number of books on the cart.
- Search: search("book cart")
- Results: Got 5 premises listing book types (history, romance, western, biographies) and a new clue: "Half the books on the bottom section were mystery books."
- Attempt: I can add the books I know: 12 + 8 + 5 + 6 = 31. But the "half mystery books" clue is a dangling premise. I can't solve the full problem yet.
**** 31

Next Action: The results were relevant, so I'll check for more with next_page().

Iteration 2
- Search: next_page() -> Results: "We have iterated through all the pages."
Analyze: That query is done. The clue mentioned a "bottom section," which implies a "top section." I need to investigate that.
- Search: search("top section")
- Results: Got a new premise: "4 poetry books from the top section."
- Attempt to Solve: Now I can piece everything together.
- Calculation:
Top section: 12 history + 8 romance + 4 poetry = 24 books.
Bottom section: The known books are 5 western + 6 biographies = 11 books. This must be one half, so the other half is 11 mystery books. Total = 11 + 11 = 22 books.
Total Cart: 24 (top) + 22 (bottom) = 46 books.
**** 46

Conclusion: All premises are used and the logic holds. I'm ready to answer.

**5.** Concluding and Final Answer
When you are confident, format your final response like this. Remember to use #### only for the final answer, not for intermediate calculations.

My calculation:
Top section: 12 history + 8 romance + 4 poetry = 24 books
Bottom section: 5 western + 6 biographies + 11 mystery = 22 books
Total: 24 + 22 = 46 books

Confidence: High
#### 46

### E.3 THE CoT PROMPT

You are a problem-solving agent. Your mission is to solve math
word problems by finding all necessary facts (premises) using
the tools provided. Your reasoning process is the most
important part of your task.

<Task>
Your job is to use tools to gather all the facts (premises)
needed to solve a math word problem. You can use any of the
tools provided to you to find the premises. Your work is
conducted in a tool-calling loop where you search for
information and then reason about it. The goal is to arrive at
a final, calculated answer based only on the premises you have
found.
</Task>

<How to Answer>
Provisional Answer: During your reasoning steps, if you
calculate a temporary answer based on incomplete information,
start the provisional answer with ****. For example:
**** 42

Final Answer: Your final, conclusive answer must begin with
#### and contain only the numerical solution. For example:
#### 42

If you feel the problem is unsolvable:
#### I don't know
</How to Answer>

<Available Tools>
You have access to these tools:

search_information: For searching your database for premises
using keywords.
next_page: Gets the next set of results for your last search
query.
</Available Tools>

<Instructions>
Think like a methodical researcher with limited time. Follow
these steps:
Read the problem carefully – What specific information do you
need to find?
Start with broader searches – Use broad, comprehensive queries
first.
After each search, pause and reason – Do I have enough facts
to solve it? What's still missing?
Execute narrower searches as you gather information – Fill in
the gaps.
Stop when you can answer confidently – Don't keep searching
unnecessarily.
</Instructions>

<Show Your Thinking>
After each search or next_page tool call, use the reasoning
tool to analyze the results:
What key information did I find?

```
What's missing?
Do I have enough to answer the question comprehensively? (Show
your calculation and provisional ** answer here if you can).
Should I search more or provide my answer? (State your next
tool call).
</Show Your Thinking>
```

### E.4 THE "THINK TOOL" PROMPT

```
You are a problem-solving agent. Your mission is to solve math
word problems by finding all necessary facts (premises) using
the tools provided. Your reasoning process is the most
important part of your task.

<Task>
Your job is to use tools to gather all the facts (premises)
needed to solve a math word problem. You can use any of the
tools provided to you to find the premises. Your work is
conducted in a tool-calling loop where you search for
information and then reason about it. The goal is to arrive at
a final, calculated answer based only on the premises you have
found.
</Task>

<How to Answer>
Provisional Answer: During your reasoning steps, if you
calculate a temporary answer based on incomplete information,
start the provisional answer with ****. For example:
**** 42

Final Answer: Your final, conclusive answer must begin with
#### and contain only the numerical solution. For example:
#### 42

If you feel the problem is unsolvable:
#### I don't know
</How to Answer>

<Available Tools>
You have access to these tools:

search_information: For searching your database for premises
using keywords.
next_page: Gets the next set of results for your last search
query.
think_tool: For reflection, calculation, and strategic
planning.

**CRITICAL: Use think_tool after each search to reflect on
results and plan next steps. Do not call think_tool with the
search_information or next_page. It should be to reflect on
the results of the search.**
</Available Tools>

<Instructions>
Think like a methodical researcher with limited time. Follow
these steps:
Read the problem carefully - What specific information do you
need to find?
```

```
Start with searches.
After each search, pause and reason.
Stop when you can answer confidently.
</Instructions>

<Show Your Thinking>
After each search or next_page tool call, use the reasoning
tool to analyze the results and plan the next steps.
</Show Your Thinking>
```

## E.5 THE "REVISIT TOOL" PROMPT

```
You are a problem-solving agent. Your mission is to solve math
word problems by finding all necessary facts (premises) using
the tools provided. Your reasoning process is the most
important part of your task.

<Task>
Your job is to use tools to gather all the facts (premises)
needed to solve a math word problem. You can use any of the
tools provided to you to find the premises. Your work is
conducted in a tool-calling loop where you search for
information and then reason about it. The goal is to arrive at
a final, calculated answer based only on the premises you have
found.
</Task>

<How to Answer>
Provisional Answer: During your reasoning steps, if you
calculate a temporary answer based on incomplete information,
start the provisional answer with ****. For example:
**** 42

Final Answer: Your final, conclusive answer must begin with
#### and contain only the numerical solution. For example:
#### 42

If you feel the problem is unsolvable:
#### I don't know
</How to Answer>

<Available Tools>
You have access to these tools:

search_information: For searching your database for premises
using keywords.
next_page: Gets the next set of results for your last search
query.
revisit: For revisiting a previous search topic with a refined
plan.

**CRITICAL: Use revisit tool if you realize you need to
revisit a previous search topic with a refined plan.**
</Available Tools>

<Active Revisit>
Use revisit when:
New info changes how you should have searched earlier.
```

A previous query was too broad, too narrow, or off-target.
You discovered a key term/structure worth a better query.
You touched a topic but didn't explore it systematically.

When calling revisit, set:
revisit_topic: the prior area to revisit.
reasoning: why returning now is better.
new_query: refined query.
</Active Revisit>

<Instructions>
Think like a methodical researcher with limited time. Follow these steps:
Read the problem carefully – What specific information do you need to find?
Start with broader searches – Use broad, comprehensive queries first.
After each search, pause and reason – Do I have enough facts to solve it? What's still missing?
Execute narrower searches as you gather information – Fill in the gaps.
Prefer revisit over aimless paging when your plan changes.
Stop when you can answer confidently – Don't keep searching unnecessarily.
</Instructions>

<Show Your Thinking>
After each search_information or next_page call, write a reasoning block that answers:
− What key information did I find?
− What's missing?
− Do I have enough to answer the question? (Show your calculation; include a provisional **** line if applicable.)
− What will I do next | call revisit, run another search/next_page, or provide my final answer? (State the next tool call explicitly, if any.)
</Show Your Thinking>

E.6   THE "EXPLORE TOOL" PROMPT

You are a problem-solving agent. Your mission is to solve math word problems by finding all necessary facts (premises) using the tools provided. Your reasoning process is the most important part of your task.

<Task>
Your job is to use tools to gather all the facts (premises) needed to solve a math word problem. You can use any of the tools provided to you to find the premises. Your work is conducted in a tool-calling loop where you search for information and then reason about it. The goal is to arrive at a final, calculated answer based only on the premises you have found.
</Task>

<How to Answer>
Provisional Answer: During your reasoning steps, if you calculate a temporary answer based on incomplete information, start the provisional answer with ****. For example:

```
**** 42

Final Answer: Your final, conclusive answer must begin with
#### and contain only the numerical solution. For example:
#### 42

If you feel the problem is unsolvable:
#### I don't know
</How to Answer>

<Available Tools>
You have access to these tools:

- Tool: search_information | returns up to five relevant
premises for a query.
- Tool: next_page | returns the next five results for the last
search.
- Tool: explore | explore a completely new research topic.
Inputs: new_explore_topic, reasoning, query.

**CRITICAL: Use explore tool if you realize you need to
explore a completely new topic.**
</Available Tools>

<Active Explore>
Use explore when:
Current approach yields limited results
You want to think from a different angle
YOu want to explore different concepts

When calling explore, set:
new_explore_topic: the related term(s) or area to explore.
reasoning: why exploring is better.
query: specific search terms for the new topic.
</Active Explore>

<Instructions>
Think like a methodical researcher with limited time. Follow
these steps:
Read the problem carefully - What specific information do you
need to find?
Start with broader searches - Use broad, comprehensive queries
first.
After each search, pause and reason - Do I have enough facts
to solve it? What's still missing?
Execute narrower searches as you gather information - Fill in
the gaps.
Prefer explore over aimless paging when your plan changes.
Stop when you can answer confidently - Don't keep searching
unnecessarily.
</Instructions>

<Show Your Thinking>
After each search_information or next_page call, write a
reasoning block that answers:
- What key information did I find?
- What's missing?
```

```
    - Do I have enough to answer the question? (Show your
    calculation; include a provisional **** line if applicable.)
    - What will I do next | call explore, run another
    search/next_page, or provide my final answer? (State the next
    tool call explicitly, if any.)
    </Show Your Thinking>
```

## F  THE USE OF LARGE LANGUAGE MODELS (LLMs)

We used LLMs mainly for grammar checking and polishing in paper writing.

