# OpenReview forum: "GSM-Agent: Understanding Agentic Reasoning Using Controllable Environments"
_ICLR.cc/2026/Conference — Submitted to ICLR 2026_

### Official Review · Reviewer_1sjM · 2025-10-27

**Soundness:** 3
**Presentation:** 3
**Contribution:** 2
**Rating:** 4
**Confidence:** 3

**Summary:**

This paper proposes a new benchmark, GSM-Agent, aimed at addressing the difficulty of existing benchmarks in isolating the evaluation of LLM "Agentic Reasoning". The benchmark is derived from GSM8K, decomposing the original problems into a "question" and multiple "premises." The agent only sees the "question" and must use tools to proactively search a database for the hidden "premise" information to solve the task.
Experiments reveal a significant gap between "static reasoning" and "agentic reasoning," with even frontier models achieving only about 67% accuracy. To analyze the reasons, the authors propose the "Agentic Reasoning Graph" framework, which maps agent behavior onto a path on the graph. A core finding is that the 'revisit' ability—returning to a previously visited node—is highly correlated with task success, and this is what many models lack. Based on this, the authors propose a "tool-augmented scaling" method, which encourages 'revisit' by adding new tools, and confirm its effectiveness.

**Strengths:**

1. **Novel and Controlled Benchmark Design:** The design of GSM-Agent is clever. It successfully decouples "agentic reasoning" (information gathering) from "complex reasoning" (the problem itself). Using grade-school math problems as a base ensures the task's inherent logical difficulty is controlled, allowing the evaluation to truly focus on the agent's information-gathering and decision-making capabilities. This ability to compare "static" vs. "agentic" settings on the same core task is a significant contribution.
2. **Insightful Analytical Framework and Key Finding:** The "Agentic Reasoning Graph" is a novel and powerful analytical tool. Discretizing the agent's trajectory into "exploration," "exploitation," and "revisit" behaviors provides clear, quantitative metrics for understanding agent behavior. This framework leads to the paper's core contribution: identifying "revisit" as a key skill for agent success. This point is strongly supported by the high correlation found between the "revisit ratio" and "accuracy".

**Weaknesses:**

1. **Generalizability:** The benchmark is built entirely on GSM8K (math problems). While this provides good control, it also limits the generalizability of the conclusions. Is the identified importance of "revisit" equally applicable to other types of agentic tasks? The paper claims to study broad "agentic reasoning," but the evidence is derived from a single domain.
2. **Sensitivity of the Analytical Framework:** The entire "revisit" conclusion is highly dependent on the construction of the "Agentic Reasoning Graph". This construction, in turn, relies on K-means clustering (with $K=250$) and a specific embedding model (`text-embedding-3-large`). The authors do not discuss the sensitivity of these hyperparameter choices. If the value of K were changed or a different embedding model were used, the granularity of the nodes would change. Would the strong correlation between "revisit ratio" and "accuracy" still hold?

**Questions:**

1. **Regarding Generalizability**: How do the authors view the generalizability of the "revisit" insight to non-mathematical, more agentic tasks? Is "revisit" still expected to be a key bottleneck in those tasks?
2. **Regarding the Robustness of the Analytical Framework:** Can the authors provide ablation studies on the construction of the "Agentic Reasoning Graph"? Specifically, how much does the strong correlation between "revisit ratio" and "accuracy" change if the number of clusters in K-means is varied or if a different document embedding model is used?

---

> ### Author Response · Authors · 2025-11-27
>
> We thank the reviewer for the helpful and insightful feedback. Below, we provide our responses.
>
> > 1. Generalizability: The benchmark is built entirely on GSM8K (math problems). While this provides good control, it also limits the generalizability of the conclusions. Is the identified importance of "revisit" equally applicable to other types of agentic tasks? The paper claims to study broad "agentic reasoning," but the evidence is derived from a single domain.
>
> We refer reviewers to our response to CQ1 in the global response on the generalizability of our discovered agentic reasoning patterns.
>
> > 2. Sensitivity of the Analytical Framework: The entire "revisit" conclusion is highly dependent on the construction of the "Agentic Reasoning Graph". This construction, in turn, relies on K-means clustering (with ) and a specific embedding model (text-embedding-3-large). The authors do not discuss the sensitivity of these hyperparameter choices. If the value of K were changed or a different embedding model were used, the granularity of the nodes would change. Would the strong correlation between "revisit ratio" and "accuracy" still hold?
>
> We thank the reviewer for the suggestion. Please see our response to CQ2 in the global response.
>
>
> > 1. Regarding Generalizability: How do the authors view the generalizability of the "revisit" insight to non-mathematical, more agentic tasks? Is "revisit" still expected to be a key bottleneck in those tasks?
>
> Please see our responses to Weakness 1.
>
> > 2. Regarding the Robustness of the Analytical Framework: Can the authors provide ablation studies on the construction of the "Agentic Reasoning Graph"? Specifically, how much does the strong correlation between "revisit ratio" and "accuracy" change if the number of clusters in K-means is varied or if a different document embedding model is used?
>
> Please see our responses to Weakness 2.

---

### Official Review · Reviewer_uRSp · 2025-10-30

**Soundness:** 2
**Presentation:** 3
**Contribution:** 2
**Rating:** 4
**Confidence:** 4

**Summary:**

This paper first proposes GSM-Agent, a benchmark designed to evaluate the agentic reasoning of LLMs. The benchmark is adapted from the GSM8K dataset, which decomposes and rewrites each original problem into an under-defined question (task) and relevant documents transformed from the problem premises. To answer the question, LLMs are supposed to use search tools to find the relevant documents from the whole dataset’s document collection. Based on GSM-Agent, this paper then defines an agentic reasoning graph to track the exploration, exploitation and revisit steps in the tool-use traces of LLM agents, based on clustering document embeddings into graph nodes and mapping document searching in the traces into paths on the nodes. Experimental analysis reveals that revisit behavior, i.e., returning to previously accessed documents, is strongly correlated with successful agentic reasoning. Based on that, this paper further proposes a tool-augmented scaling method to encourage the revisit behavior and thus improve reasoning accuracy of LLM agents.

**Strengths:**

- GSM-Agent provides a controllable testbed for grade school math reasoning, which enables the evaluation to switch between fully-defined question answering and under-defined agentic task solving, which is not covered by prior GSM benchmarks.
- The experimental analysis based on the agentic reasoning graph is insightful, especially, the finding of the strong correlation between revisit behavior and agentic reasoning accuracy has valuable potential to benefit future design of LLM agentic frameworks.
- The experimental study spans a comprehensive set of LLM agents, which avoids model-specific conclusions.

**Weaknesses:**

- The study is limited to a single GSM domain, which makes it questionable whether the proposed findings and methods could generalize to other broader domains. In particular, with respect to GSM-Agent, similar benchmarks that support retrieval-based reasoning have long existed, namely HotpotQA [1] and MuSiQue [2] based on Wikipedia documents. Controllable experiments could also be done by either plugging in a Wikipedia search engine or directly giving the relevant Wikipedia documents to LLM agents.
- It is unclear how the defined agentic reasoning graph is different from or improved upon the related work of reasoning topology (Minegishi et al., 2025) cited in the paper, which limits the novelty of the proposed analytical methodology of agentic reasoning.
- The study overlooks the clarification of confounders that may affect agentic reasoning accuracy. For example, after encouraging more proportion of revisit steps, it is unclear whether the improvement is due to rechecking previous relevant documents to trigger better reflection, or just because of accessing less kinds of irrelevant documents in the top-k retrieval, thus being less distracted but not performing better thinking.

[1] HotpotQA: A Dataset for Diverse, Explainable Multi-hop Question Answering. Yang et al., 2018.

[2] MuSiQue: Multihop Questions via Single-hop Question Composition. Trivedi et al., 2022.

**Questions:**

- In Table 1, why is the search-complete (SC) rate, i.e., the proportion of tasks that an agent finds all relevant documents, always lower than the final answer accuracy? Does this mean that the agent often correctly guesses the answer even if the relevant documents (premises) are not fully found? If so, the results of accuracy may not be reliable enough to measure the agentic reasoning performance.
- In section 3.2.4, should 32315 be the number of unique documents (premises) in the database instead of unique questions? This may be a typo.

---

> ### Author Response · Authors · 2025-11-27
>
> We thank the reviewer for the helpful and insightful feedback. Below, we provide our responses.
>
> > The study is limited to a single GSM domain, which makes it questionable whether the proposed findings and methods could generalize to other broader domains. In particular, with respect to GSM-Agent, similar benchmarks that support retrieval-based reasoning have long existed, namely HotpotQA [1] and MuSiQue [2] based on Wikipedia documents. Controllable experiments could also be done by either plugging in a Wikipedia search engine or directly giving the relevant Wikipedia documents to LLM agents.
>
> We argue that the tasks in HotpotQA and MuSiQue would be easier than our GSM-Agent benchmark in the agentic setting since it is easy to infer “what to ask” directly from the question in HotpotQA and MuSiQue. For example, for the question “What was the former band of the member of Mother Love Bone who died just before the release of ‘Apple’?” in the Hotpot QA, it is pretty straightforward to know what to ask (i.e., first search “the member of Mother Love Bone who died just before the release of ‘Apple’”, get the answer “Andrew Wood” from the retrieved document and then search “the former band of Andrew Wood”). However, in our benchmark, it is not easy to identify the correct question to ask, and models need to reason about it or even determine what to ask after retrieving several documents, or revisit previously explored searches. Therefore, our benchmark provides unique challenges for LLMs compared to (the agentic version of) existing benchmarks such as HotpotQA and MuSiQue.
>
> However, we agree with the reviewer that experimental results on how our findings and proposed tool-augmented scaling methods can generalize to broader domains can strengthen our results. Please see our additional experimental results in CQ1 in the global response.
>
> > It is unclear how the defined agentic reasoning graph is different from or improved upon the related work of reasoning topology (Minegishi et al., 2025) cited in the paper, which limits the novelty of the proposed analytical methodology of agentic reasoning.
>
> Our proposed agentic reasoning graph is inspired by the work of reasoning topology (Minegishi et al., 2025), and our work shares the basic concepts, such as the graph, nodes, and path, with Minegishi et al., 2025. However, the most important contribution in our framework is how to define the graph in agentic settings to discover important agentic reasoning patterns. The most important part of defining the graph is to define the nodes (i.e., their representations), which requires non-trivial and careful design. While Minegishi et al. (2025) use internal hidden states as node representation for static reasoning, our work identifies the appropriate granularity of the agentic reasoning step and proposes to use embedding of the search and document as node and path representations. Moreover, while the framework of Minegishi et al. (2025) is helpful in discovering the reasoning pattern in static settings, their results are at an embedding-level and cannot be directly used to improve the model’s reasoning capability since it is hard to control the hidden state directly. In contrast, our finding of the revisit pattern is behavior-level granularity and thus can be used to compare across different models and can be directly transferred to a tool-augmented test-time scaling method that improves models’ agentic reasoning capability. This not only shows the novelty of our proposed framework but also its practicality. We also believe our framework can serve as a clean, useful tool for analyzing agentic reasoning patterns in future research.
>
> > The study overlooks the clarification of confounders that may affect agentic reasoning accuracy. For example, after encouraging more proportion of revisit steps, it is unclear whether the improvement is due to rechecking previous relevant documents to trigger better reflection, or just because of accessing less kinds of irrelevant documents in the top-k retrieval, thus being less distracted but not performing better thinking.
>
> To show that the performance improvement is indeed due to the revisit capability rather than other possible confounders, such as accessing fewer irrelevant documents, we provide additional experimental results on the effectiveness of our proposed tool-augmenting test-time scaling methods in other tasks. In particular, see our responses to CQ1 in the global response.

---

> > ### Author Response · Authors · 2025-11-27
> >
> > > In Table 1, why is the search-complete (SC) rate, i.e., the proportion of tasks that an agent finds all relevant documents, always lower than the final answer accuracy? Does this mean that the agent often correctly guesses the answer even if the relevant documents (premises) are not fully found? If so, the results of accuracy may not be reliable enough to measure the agentic reasoning performance.
> >
> > The reason is that sometimes it is possible for a document to contain information about multiple premises, and thus it is possible for an agent to gather all the information even if it does not find all the documents corresponding to that question. The agent is not able to guess the answer if they did not gather the necessary information, and thus, the accuracy is reliable. We will state it clearly in the revision.
> >
> > > In section 3.2.4, should 32315 be the number of unique documents (premises) in the database instead of unique questions? This may be a typo.
> >
> > Yes, it should be “documents” instead of “problems”. Thanks for pointing this out!

---

### Official Review · Reviewer_FXj6 · 2025-10-31

**Soundness:** 2
**Presentation:** 2
**Contribution:** 2
**Rating:** 2
**Confidence:** 4

**Summary:**

This paper introduces GSM-Agent, a benchmark designed to evaluate and analyze agentic reasoning—the ability of LLMs to reason while interacting with external tools—under a clean and controllable setup.
The authors adapt the GSM8K dataset into a simulated environment where models must actively retrieve missing premises using search tools before solving math problems. The benchmark allows precise control of difficulty via distractor documents and provides a consistent comparison between static and agentic reasoning. To interpret agent behavior, the paper proposes the Agentic Reasoning Graph (ARG), which maps search traces into discrete nodes representing clusters of environment embeddings. Analysis reveals that the revisit ratio (returning to a previously explored node) strongly correlates with task accuracy. Finally, a tool-augmented scaling method introduces explicit revisit tools to encourage iterative reasoning, improving model performance compared with naive interaction scaling.

**Strengths:**

1. Well-constructed benchmark. The GSM-Agent dataset offers a rare controlled setup where agentic reasoning can be compared apples-to-apples with static reasoning on identical base problems.
2. Quantitative interpretability. The Agentic Reasoning Graph provides a useful formalism to visualize reasoning dynamics and correlate graph-level metrics (explore, exploit, revisit) with performance.
3. Insightful empirical finding. The identification of revisit behavior as a core agentic skill offers an interesting direction for future reasoning-oriented architectures.

**Weaknesses:**

1. The benchmark mainly extends GSM8K into a retrieval environment. While the controllability is commendable, the core idea—transforming reasoning problems into search tasks—is incremental relative to prior works such as AppWorld, ToolSandbox, and τ-Bench. The “agentic reasoning graph” is conceptually borrowed from earlier reasoning topology analyses (e.g., Minegishi et al., 2025) with minimal methodological novelty.

2. A large portion of the paper describes data processing (entity renaming, timestamping, anonymization), which, while meticulous, dilutes the conceptual contribution. The central analytical claim—the link between revisit ratio and accuracy—could have been explored more deeply or theoretically justified.

3. The benchmark is fully built from grade-school math problems, limiting its applicability to broader agentic domains (e.g., open-domain QA, code synthesis, multimodal planning). The paper does not test whether the identified patterns generalize beyond arithmetic reasoning.

4. While the Agentic Reasoning Graph provides an interpretable diagnostic, it lacks formal rigor—no quantitative justification is given for why the revisit ratio should causally imply stronger reasoning, or how it might relate to planning-oriented cognitive models.

5. Insufficient ablations and reliability checks.
   The paper lacks:
   * Statistical validation of correlations (e.g., confidence intervals for (R^2 = 0.914) in Fig. 4).
   * Sensitivity analysis for the chosen clustering hyperparameters (e.g., K = 250).
   * Verification of annotation or tool-use consistency across random seeds.

6. The authors acknowledge that adding “revisit tools” improves performance for weaker models, but the improvement magnitude is small and inconsistent (Fig. 5). The work does not discuss why some models (e.g., GPT-5, o3) barely benefit—suggesting the method might not scale universally.

7. Since all results are collected via LangChain-based calls, the paper may confound *prompt engineering* effects with genuine reasoning ability. The evaluation framework’s reproducibility under open-source conditions remains uncertain.

**Questions:**

1. Please report statistical significance (e.g., Pearson/Spearman coefficients with p-values) for the correlation between revisit ratio and accuracy (Fig. 4b). The current (R^2) alone does not confirm robustness.
2. The Agentic Reasoning Graph uses (K = 250) clusters. Please include ablations for different K values (e.g., 100, 500) to test stability of exploration/exploitation/revisit metrics.
3. Extend GSM-Agent to non-math tasks (e.g., commonsense QA or simple retrieval puzzles) to test whether the “revisit hypothesis” generalizes beyond arithmetic reasoning.
4. Report whether the “Revisit” tool improves reasoning in unseen environments (e.g., new document distributions). If gains vanish, it would suggest overfitting to the current dataset.
5. Provide qualitative examples of failed reasoning traces with low revisit ratios—do these failures stem from early search termination, irrelevant document focus, or token limit exhaustion?
6. Include open-source code or pseudo-code for your LangChain evaluation pipeline to ensure replicability.
7. A fair baseline against ToolSandbox (Lu et al., 2024) or AppWorld (Trivedi et al., 2024) would strengthen the claim that GSM-Agent uniquely isolates agentic reasoning effects.

---

> ### Author Response · Authors · 2025-11-27
>
> We thank the reviewer for the helpful and insightful feedback. Below, we provide our responses.
>
> > 1. The benchmark mainly extends GSM8K into a retrieval environment. While the controllability is commendable, the core idea—transforming reasoning problems into search tasks—is incremental relative to prior works such as AppWorld, ToolSandbox, and τ-Bench. The “agentic reasoning graph” is conceptually borrowed from earlier reasoning topology analyses (e.g., Minegishi et al., 2025) with minimal methodological novelty.
>
> **Comparison to prior works:**
>
> First, previous benchmarks (e.g., AppWorld, ToolSandbox, and τ-Bench) focus on more general agentic ability, while our work focuses on models' capability of combining tool-use and mathematical reasoning.
>
> Second, our benchmark provides an apple-to-apple comparison of static and agentic reasoning ability, while previous benchmarks couple different advanced capabilities and make it relatively hard to analyze.
>
> Third, our work not only proposes a benchmark but also introduces the agentic reasoning graph, which is a novel and powerful quantitative tool to facilitate analysis of agentic reasoning patterns and results in novel tool-augmented test-time scaling methods to improve models’ performance.
>
> **Novelty of agentic reasoning graph:**
>
> Our proposed agentic reasoning graph is inspired by the work of reasoning topology (Minegishi et al., 2025), and our work shares the basic concepts, such as the graph, nodes, and path, with Minegishi et al., 2025. However, the most important contribution in our framework is how to define the graph in agentic settings to discover important agentic reasoning patterns. The most important part of defining the graph is to define the nodes (i.e., their representations), which requires non-trivial and careful design. While Minegishi et al. (2025) use internal hidden states as node representation for static reasoning, our work identifies the appropriate granularity of the agentic reasoning step and proposes to use embedding of the search and document as node and path representations. Moreover, while the framework of Minegishi et al. (2025) is helpful in discovering the reasoning pattern in static settings, their results are at an embedding-level and cannot be directly used to improve the model’s reasoning capability since it is hard to control the hidden state directly. In contrast, our finding of the revisit pattern is behavior-level granularity and thus can be used to compare across different models and can be directly transferred to a tool-augmented test-time scaling method that improves models’ agentic reasoning capability. This not only shows the novelty of our proposed framework but also its practicality. We also believe our framework can serve as a clean, useful tool for analyzing agentic reasoning patterns in future research.
>
> > 2. A large portion of the paper describes data processing (entity renaming, timestamping, anonymization), which, while meticulous, dilutes the conceptual contribution. The central analytical claim—the link between revisit ratio and accuracy—could have been explored more deeply or theoretically justified.
>
> Thanks for your suggestion. We uploaded a revision where we added more analytical claim in the main text in the second half of the paper.
>
> > 3. The benchmark is fully built from grade-school math problems, limiting its applicability to broader agentic domains ...
>
> We thank the reviewer for the great suggestion. To test the identified patterns (i.e., the revisit capability is important for agentic reasoning capability), we conducted additional experiments to show its generality. Please see our response to CQ1 in the global response.
>
> > 4. While the Agentic Reasoning Graph provides an interpretable diagnostic, it lacks formal rigor—no quantitative justification is given for why the revisit ratio should causally imply stronger reasoning, or how it might relate to planning-oriented cognitive models.
>
> We agree that further analysis of the revisit pattern is an important next step, and we believe our contribution of identifying this pattern is already significant, and further analysis can be left as future work. We also provided an example of the trajectory that benefits from the revisit pattern in Appendix C.4 in our revised pdf to serve as qualitative evidence.
>
> > 5. Insufficient ablations and reliability checks.
>
> For the statistical validation of correlations and sensitivity analysis for the chosen clustering hyperparameters, we conducted additional experiments to show the robustness of our findings. See details in our response to CQ2 in the global response.
>
> For random seeds, note that we already used three different random seeds during our experiments, and the results are consistent across different seeds. For example, in the caption of Table 1, we stated that “All results are averaged over three random seeds“. We will make this clearer.

---

> > ### Author Response · Authors · 2025-11-27
> >
> > > 6. The authors acknowledge that adding “revisit tools” improves performance for weaker models, but the improvement magnitude is small and inconsistent (Fig. 5). The work does not discuss why some models (e.g., GPT-5, o3) barely benefit—suggesting the method might not scale universally.
> >
> > Our conjecture is that, for strong models such as GPT-5 and o3, the bottleneck is not revisit capability, since they already achieve relatively high performance. For those SOTA models, further improvement of the performance on our benchmark might require the improvement of other advanced capabilities, which are beyond the scope of our work. In our paper, we focus on why the tested models can all achieve perfect performance in static settings but exhibit a large gap in agentic settings, and we identify that this highly correlates with revisit capability. Our experiments also showed that encouraging revisit indeed closes the gap between strong and weak models.
> >
> > > 7. Since all results are collected via LangChain-based calls, the paper may confound prompt engineering effects with genuine reasoning ability. The evaluation framework’s reproducibility under open-source conditions remains uncertain.
> > We first argue that LangChain is a widely used pipeline, and our prompt for the main results uses zero-shot prompting, thereby eliminating confound prompt engineering effects as much as possible.
> >
> > We also agree with the reviewer that prompt engineering effects might affect the genuine reasoning ability, so we reported results using different prompt strategies in Table 4 in the appendix (such as few-shot and CoT). We provided all the evaluation prompts in Appendix E.
> >
> > Our code for the evaluation framework is provided in the supplementary materials to ensure reproducibility.
> >
> > > 1. Please report statistical significance (e.g., Pearson/Spearman coefficients with p-values) for the correlation between revisit ratio and accuracy (Fig. 4b). The current (R^2) alone does not confirm robustness.
> >
> > Please see our responses to CQ2 in the global response, which provide the p-value and further show the strong correlation between the revisit ratio and accuracy.
> >
> > > 2. The Agentic Reasoning Graph uses (K = 250) clusters. Please include ablations for different K values (e.g., 100, 500) to test stability of exploration/exploitation/revisit metrics.
> >
> > Please see our responses to CQ2 in the global response on the ablations for different K values.
> >
> > > 3. Extend GSM-Agent to non-math tasks (e.g., commonsense QA or simple retrieval puzzles) to test whether the “revisit hypothesis” generalizes beyond arithmetic reasoning.
> >
> > Please see our responses to CQ1 in the global response, which shows the revisit hypothesis generalizes beyond arithmetic reasoning.
> >
> > > 4. Report whether the “Revisit” tool improves reasoning in unseen environments (e.g., new document distributions). If gains vanish, it would suggest overfitting to the current dataset.
> >
> > We emphasize that the “Revisit” tool is independent of the environment. It just encourages the model to revisit previously explored documents/queries, which is a general pattern and does not depend on the concrete environment.  Therefore, the environments we used in our paper are actually unseen to the tool, and thus, there is no overfitting to the current dataset.
> >
> > > 5. Provide qualitative examples of failed reasoning traces with low revisit ratios—do these failures stem from early search termination, irrelevant document focus, or token limit exhaustion?
> >
> > We included an example of a reasoning trace in Appendix C4 using o3 with zero-shot prompting. Note that before Step 30, the agent fails to solve the problem since it cannot find new useful documents. The key step is Step 31, where the agent revisits Node 9 with the keyword “coupon” in the query and found two new important documents. After that, the agent successfully solves the task within just six steps.
> >
> > > 6. Include open-source code or pseudo-code for your LangChain evaluation pipeline to ensure replicability.
> >
> > Our code for the evaluation framework is provided in the supplementary materials, and our evaluation prompt is provided in Appendix E to ensure reproducibility.
> >
> > > 7. A fair baseline against ToolSandbox (Lu et al., 2024) or AppWorld (Trivedi et al., 2024) would strengthen the claim that GSM-Agent uniquely isolates agentic reasoning effects.
> >
> > We argue that it is hard to isolate the agentic reasoning effect for previous work, such as ToolSandbox or AppWorld, due to their complex nature in the task. Indeed, this shows the unique advantage of our benchmark that it provides a clean and controllable environment to isolate agentic reasoning effects.

---

### Official Review · Reviewer_v5zn · 2025-11-01

**Soundness:** 3
**Presentation:** 3
**Contribution:** 3
**Rating:** 4
**Confidence:** 4

**Summary:**

This work:
1) builds the GSM-Agent benchmark to isolate agentic reasoning (by requiring LLMs to collect necessary premises via tools for grade-school-level problems, avoiding mixing it with other advanced capabilities);
2) proposes the agentic reasoning graph to analyze reasoning patterns and identify that many models lack the "revisit" ability critical for agentic reasoning;
3) develops a tool-augmented test-time scaling method to improve LLMs’ agentic reasoning performance by encouraging revisit.

**Strengths:**

1. GSM-Agent’s controllable environment enables unique static-agentic reasoning comparisons by isolating agentic skills via simple math, avoiding confounding factors like advanced expertise.
2. The agentic reasoning graph provides a reproducible framework to quantify exploration/exploitation/revisit, with strong data $R^2=0.914$ validating revisit as a core skill.
3. The tool-augmented method outperforms naive interaction scaling, offering a practical, low-cost way to boost agentic reasoning without retraining.
4. Comprehensive evaluation of 12 models across metrics gives a holistic view of agentic performance.

**Weaknesses:**

1. The choice of K=250 clusters for the agentic reasoning graph lacks justification or ablation, weakening the graph’s robustness.
2. Missing critical baselines (e.g., perfect retrieval) and tool ablations (e.g., Revisit(·) alone) make it hard to rule out confounding factors.
3. No evidence that revisit’s importance generalizes beyond grade-school math, limiting claims about its role as a general agentic skill.
4. "The performance of GPT-5 stands at only 67%". Yet in my view, this is a relatively high score when considering the benchmark’s complexity. For context, take OSWorld in another domain: when this benchmark was first proposed, a mere 30% performance rate was already regarded as high.

**Questions:**

Address the weaknesses.

---

> ### Author Response · Authors · 2025-11-27
>
> We thank the reviewer for the helpful and insightful feedback. Below, we provide our responses.
>
> > The choice of K=250 clusters for the agentic reasoning graph lacks justification or ablation, weakening the graph’s robustness.
>
> We thank the reviewer for the suggestion. Please see our response to CQ2 in the global response.
>
> > Missing critical baselines (e.g., perfect retrieval) and tool ablations (e.g., Revisit(·) alone) make it hard to rule out confounding factors.
>
> The “perfect retrieval” baseline is equivalent to directly providing all the relevant documents in the prompt. Note that in our dataset construction procedure, we already used Claude-3.5 as a filter to retain all tasks solvable by Claude-3.5 when the prompt provides full information. Therefore, all tasks in our GSM-agent benchmark can be solved with perfect retrieval.
>
> Also, note that we already conducted tool ablations in Section 4.3. We emphasize that the search() and nextpage() tools are two basic tools in our settings. A toolset with Revisit() only, without search() or Nextpage(), is not well-defined, since the agent must search before revisiting. Therefore, in the four ablation combinations of tools in Section 4.3, we keep search() and nextpage() while varying the additional tool combinations.
>
> > No evidence that revisit’s importance generalizes beyond grade-school math, limiting claims about its role as a general agentic skill.
>
> We refer reviewers to our response to CQ1 in the global response.
>
> > "The performance of GPT-5 stands at only 67%". Yet in my view, this is a relatively high score when considering the benchmark’s complexity. For context, take OSWorld in another domain: when this benchmark was first proposed, a mere 30% performance rate was already regarded as high.
>
> We agree that a 67% performance is relatively high. We said ``only’’ because we are comparing the agentic setting with the sciatic setting, where GPT-5 can achieve perfect performance. The ~30% performance drop is considered high (and can be as high as 80%-90% for other models), as the core problem is still a grade-school math problem.

---

### Author Response · Authors · 2025-11-27
**Global Response (1)**

We thank the reviewers for their time and effort in reviewing our paper and providing insightful and helpful feedback. Below, we first address several common questions (CQs).

### CQ1: Generalizability of our discovered ``revisit’’ reasoning skill beyond our GSM-agent benchmark.

Answer:

We thank the reviewers for the great suggestions. To show that revisit capability is an important skill in more general agentic settings beyond our proposed GSM-Agent, we further evaluate it on DeepResearch Bench [1], which is a comprehensive benchmark for deep research agents consisting of 100 PhD-level research tasks covering 22 real-world fields such as science and technology, business and finance, and software. We follow the official evaluation protocol and report RACE scores on this benchmark, and summarize our main results in the following table. (See details in Table 3 and Section 4.4 in our updated version).

| Agents | Overall Score (p-value) |
|-------------------------|-------------------|
| GPT-4.1 langchain w/ revisit      | 43.85\% ($p \approx 0.01$) |
|GPT-4.1 langchain w/o revisit         | 43.33\%     |
| GPT-5 langchain w/ revisit      | 49.59\% ($p \approx 0.5$) |
|GPT-5 langchain w/o revisit         | 49.70\%     |
| GPT-5 revisit only      | 51.44\% ($p \approx $1e-17) |

We inject revisit behavior into the deep research agent in two ways. First, we augment the LangChain Open Deep Research multi-agent supervisor architecture with an explicit revisit tool available to the supervisor agent. As in our analysis in Section 4.3, the supervisor periodically reviews the history of search queries, selects a single under-explored topic, and invokes the revisit tool to issue focused follow-up searches on that topic. For a fair comparison, we keep the total number of research iterations and the degree of concurrency identical to the original Open Deep Research configuration, so that the presence of the revisit tool is the only design difference. Second, we construct a revisit-only agent by removing the original langchain multi-agent design (summary, search workers, and supervisor) and retaining a single agent equipped solely with the revisit tool. In this variant, the main research loop degenerates into a single agent that iteratively runs search-and-revise cycles on the current draft report, using revisit-structured searches to refine the analysis.

The above table shows that, for both GPT-5 and GPT-4.1 within the LangChain framework, adding the revisit tool consistently improves performance or maintains comparable performance on the DeepResearch Bench. Although the absolute gains are small, paired t-tests indicate that these improvements are statistically reliable (for GPT-4.1, $p\approx 0.01$). Given that the gap between the Rank 1 and Rank 2 entries on the public leaderboard is only 0.01\%, such sub-percentage improvements are nontrivial at the resolution of this benchmark. We also observed that for GPT-5, adding the revisit tool does not further improve performance, consistent with results in our GSM-agent benchmark: stronger models benefit less from the revisit augmenting method, since they might already have strong revisit capability or require more advanced methods to further improve it.

Moreover, the single-agent GPT-5 revisit only configuration achieves the highest overall score in the above table, surpassing the more complex LangChain-based multi-agent systems. This suggests that explicitly modeling revisit behavior is itself a powerful design axis, and that strengthening structured revisit within a single strong model can be at least as effective as, and sometimes preferable to, introducing additional agents and orchestration. These findings prompt us to reconsider standard deep-research agent designs and highlight revisit capability as a first-class component rather than a minor implementation detail.

We kindly refer reviewers to Section 4.4 for more details and tables.

---

> ### Author Response · Authors · 2025-11-27
> **Global Response (2)**
>
> ### CQ2: Choices of hyperparameters for the agentic reasoning graph.
>
> To show that the reasoning pattern discovered in the agentic reasoning graph does not depend on the choice of hyperparameters (i.e., K = 250), we performed ablations across different values of K. Specifically, we choose  K=150, 200, 300, 350 and present the result in the following tables as well as in Appendix C4. In particular, we fit linear regressions of accuracy on the three reasoning pattern ratios and report the estimated slope, R2, and p-value for each model. The results across different values of $K$ consistently show that the model accuracy has a weak correlation to the exploration ratio, a strong correlation to the revisit ratio, and a negative correlation to the exploitation ratio.
>
>
>
> | K=150 | Exploration |  Revisit | Exploitation |
> |------------|-----------|-----------|-----------|
> | $R\^2$      | 0.049     |  0.795 |  0.345 |
> | slope         | 0.316     | 2.046    | -0.638 |
> | p-value      | 5.12e-1    | 2.26e-4 | 5.74e-2 |
>
> | K=200 | Exploration |  Revisit | Exploitation |
> |------------|-----------|-----------|-----------|
> | $R\^2$      | 0.031     |  0.884 |  0.361 |
> | slope         | 0.277     | 1.851    | -0.704 |
> | p-value      | 6.08e-1    | 1.69e-5 | 5.07e-2 |
>
> | K=250 | Exploration |  Revisit | Exploitation |
> |------------|-----------|-----------|-----------|
> | $R\^2$      | 0.038     |  0.900 |  0.330 |
> | slope         | 0.280     | 2.007    | -0.630 |
> | p-value      | 5.67e-1    | 8.46e-6 | 6.43e-2 |
>
>
> | K=300 | Exploration |  Revisit | Exploitation |
> |------------|-----------|-----------|-----------|
> | $R\^2$      | 0.028    |  0.935 |  0.290 |
> | slope         | 0.238     | 2.149    | -0.598 |
> | p-value      | 6.24e-1    | 1.24e-6 | 8.73e-2 |
>
>
> | K=350 | Exploration |  Revisit | Exploitation |
> |------------|-----------|-----------|-----------|
> | $R\^2$      | 0.028    |  0.850 |  0.316 |
> | slope         | 0.238     | 2.021    | -0.633 |
> | p-value      | 6.26e-1    | 5.44e-5 | 7.21e-2 |
>
> We kindly refer reviewers to Appendix C4 for more details and figures.
>
> **Reference:**
>
>
> [1] Du, Mingxuan, Benfeng Xu, Chiwei Zhu, Xiaorui Wang, and Zhendong Mao. "DeepResearch Bench: A Comprehensive Benchmark for Deep Research Agents." arXiv preprint arXiv:2506.11763 (2025).

---

### Meta-Review · Area_Chair_ExNi · 2026-01-07

**Summary:**

The paper introduces GSM-Agent, a benchmark designed to isolate "agentic reasoning" by requiring models to retrieve missing premises for grade-school math problems via tool use. To analyze agent behavior, the authors propose the Agentic Reasoning Graph (ARG), which clusters environment interactions into nodes. Their key finding is that the "revisit ratio" (the frequency with which an agent returns to previously explored information) is a high-signal predictor of success. The reviewers initially expressed concerns regarding the technical robustness of the ARG clustering (specifically the choice of $K=250$), the generalizability of the "revisit" finding beyond simple math tasks, and the incremental nature of the benchmark relative to existing retrieval-based reasoning datasets.

**Reviewer Concerns:**

Addressed by Rebuttal:

**Robustness of ARG** ($K$ hyperparameter): The authors provided extensive sensitivity analysis across $K \in \{150, 200, 300, 350\}$. The results consistently show high $R^2$ values and significant $p$-values, confirming that the correlation between revisit ratio and accuracy is not an artifact of the clustering granularity.

**Generalizability beyond GSM**: A major concern was whether "revisiting" was unique to arithmetic search. The authors' evaluation on the DeepResearch Bench (PhD-level tasks) demonstrated that adding a revisit tool improved performance for GPT-4.1 and reached SOTA levels in single-agent GPT-5 configurations. This significantly strengthens the paper's claims.

**Statistical Rigor**: The authors provided the requested Pearson/Spearman coefficients and $p$-values (e.g., $p < 1e-5$ for revisit correlations), moving the work beyond purely qualitative observation.Baseline Clarifications: The authors clarified the "perfect retrieval" baseline and provided tool-use ablations that were previously misunderstood.

Not yet solved:

**Causality vs. Correlation**: While the authors provided qualitative traces to show how revisiting leads to success, a formal theoretical or causal proof of why "revisit" implies stronger reasoning remains for future work.

**Marginal Gains for SOTA Models**: Reviewers noted that GPT-5 and o3 see diminishing returns from the explicit "revisit tool." While the authors hypothesize this is because these models have internalised the skill, the method's utility for the most advanced models is less pronounced than for the "mid-tier" models.

**Comparison to Prior Benchmarks**: Some reviewers (FXj6) may still view the benchmark as incremental relative to AppWorld or ToolSandbox, though the authors have argued well for the "cleanliness" and "static-vs-agentic" control GSM-Agent offers.

**Reviewer Scores:**

v5zn might increase the score.

---

### Decision · Program_Chairs · 2026-01-26

Reject